# Exploring Memorization in Adversarial Training

**Yinpeng Dong**[1,2], **Ke Xu**[4], **Xiao Yang**[1], **Tianyu Pang**[1], **Zhijie Deng**[1], **Hang Su**[1,3], **Jun Zhu**[1,2,3]*

[1] Dept. of Comp. Sci. and Tech., Institute for AI, Tsinghua-Bosch Joint ML Center, THBI Lab
[1] BNRist Center, Tsinghua University, Beijing, China; [2] RealAI; [3] Peng Cheng Laboratory; [4] CMU

{dongyinpeng, suhangss, dcszj}@mail.tsinghua.edu.cn, kx1@andrew.cmu.edu

## Abstract

Deep learning models have a propensity for fitting the entire training set even with random labels, which requires *memorization* of every training sample. In this paper, we explore the memorization effect in adversarial training (AT) for promoting a deeper understanding of model capacity, convergence, generalization, and especially robust overfitting of the adversarially trained models. We first demonstrate that deep networks have sufficient capacity to memorize adversarial examples of training data with completely random labels, but *not* all AT algorithms can converge under the extreme circumstance. Our study of AT with random labels motivates further analyses on the convergence and generalization of AT. We find that some AT approaches suffer from a gradient instability issue and most recently suggested complexity measures cannot explain robust generalization by considering models trained on random labels. Furthermore, we identify a significant drawback of memorization in AT that it could result in robust overfitting. We then propose a new mitigation algorithm motivated by detailed memorization analyses. Extensive experiments on various datasets validate the effectiveness of the proposed method.

## 1 Introduction

Deep neural networks (DNNs) usually exhibit excellent generalization ability in pattern recognition tasks, despite their sufficient capacity to overfit or *memorize* the entire training set with completely random labels (Zhang et al., 2017). The memorization behavior in deep learning has aroused tremendous attention to identifying the differences between learning on true and random labels (Arpit et al., 2017; Neyshabur et al., 2017), and examining what and why DNNs memorize (Feldman, 2020; Feldman & Zhang, 2020; Maennel et al., 2020). This phenomenon has also motivated a growing body of works on model capacity (Arpit et al., 2017; Belkin et al., 2019), convergence (Allen-Zhu et al., 2019; Du et al., 2019; Zou et al., 2020), and generalization (Neyshabur et al., 2017; Bartlett et al., 2017), which consequently provide a better understanding of the DNN working mechanism.

In this paper, we explore the memorization behavior for a different learning algorithm—**adversarial training (AT)**. Owing to the security threat of adversarial examples, i.e., maliciously generated inputs by adding imperceptible perturbations to cause misclassification (Szegedy et al., 2014; Goodfellow et al., 2015), various defense methods have been proposed to improve the adversarial robustness of DNNs (Kurakin et al., 2017; Madry et al., 2018; Liao et al., 2018; Wong & Kolter, 2018; Cohen et al., 2019; Zhang et al., 2019b; Pang et al., 2019; 2020; Dong et al., 2020a). AT is arguably the most effective defense technique (Athalye et al., 2018; Dong et al., 2020b), in which the network is trained on the adversarially augmented samples instead of the natural ones (Madry et al., 2018).

Despite the popularity, the memorization behavior in AT is less explored. Schmidt et al. (2018) show that a model is able to fully (over)fit the training set against an adversary, i.e., reaching almost $100\%$ robust training accuracy, while the performance on test data is much inferior, witnessing a significant generalization gap. The overfitting phenomenon in AT is further investigated in Rice et al. (2020). However, it is not clear *whether DNNs could memorize adversarial examples of training data with completely random labels*. Answering this question could help to examine the effects of memorization in AT under the "extreme" circumstance and facilitate a deeper understanding of capacity, convergence, generalization, and robust overfitting of the adversarially trained models. In general, it

---

*Jun Zhu is the corresponding author. This work was done when Ke Xu was visiting Tsinghua University.

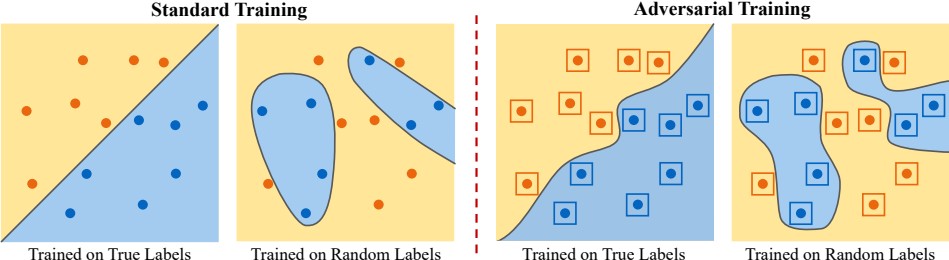

Figure 1: A conceptual illustration of decision boundaries learned via standard training and adversarial training with true and random labels, respectively. The model needs a significantly more complicated decision boundary to memorize adversarial examples of training data with random labels.

is difficult for a classifier to memorize adversarial examples with random labels since the model entails a much more complicated decision boundary, as illustrated in Fig. 1. Even though the networks have sufficient capacity, AT may not necessarily converge. Therefore, we aim to comprehensively study this problem and explore how the analysis can motivate better algorithms.

**Our contributions.** We first empirically investigate the memorization behavior in AT by performing **PGD-AT** (Madry et al., 2018) and **TRADES** (Zhang et al., 2019b) with random labels sampled uniformly over all classes. Different from standard training (ST) that can easily memorize random labels (Zhang et al., 2017), AT may *fail to converge*, with PGD-AT being a typical example. Nevertheless, TRADES can converge under this circumstance. It demonstrates that DNNs have *sufficient capacity* to memorize adversarial examples of training data with completely random labels. This phenomenon is commonly observed on multiple datasets, network architectures, and threat models.

The memorization analysis has further implications for understanding the *convergence* and *generalization* of AT. We conduct a convergence analysis on gradient magnitude and stability to explain the counter-intuitive different convergence properties of PGD-AT and TRADES with random labels since they behave similarly when trained on true labels (Rice et al., 2020). We corroborate that PGD-AT suffers from a gradient instability issue while the gradients of TRADES are relatively stable thanks to its adversarial loss. Moreover, by considering models trained on random labels, our generalization analysis indicates that several recently suggested complexity measures are inadequate to explain robust generalization, which is complementary to the findings in ST (Neyshabur et al., 2017). Accordingly, an appropriate explanation of robust generalization remains largely under-addressed.

Lastly, but most importantly, we identify a significant drawback of memorization in AT that it could result in *robust overfitting* (Rice et al., 2020). We argue that the cause of robust overfitting lies in the memorization of one-hot labels in the typical AT methods. The one-hot labels can be inappropriate or even *noisy* for some adversarial examples because some data naturally lies close to the decision boundary, and the corresponding adversarial examples should be assigned low predictive confidence (Stutz et al., 2020; Cheng et al., 2020). To solve this problem, we propose a new mitigation algorithm that impedes over-confident predictions by regularization for avoiding the excessive memorization of adversarial examples with possibly noisy labels. Experiments validate that our method can eliminate robust overfitting to a large extent across multiple datasets, network architectures, threat models, and AT methods, achieving better robustness under a variety of adversarial attacks than the baselines.

## 2 BACKGROUND

### 2.1 ADVERSARIAL TRAINING

Let $\mathcal{D} = \{(\mathbf{x}_i, y_i)\}_{i=1}^n$ denote a training dataset with $n$ samples, where $\mathbf{x}_i \in \mathbb{R}^d$ is a natural example and $y_i \in \{1, ..., C\}$ is its true label often encoded as an one-hot vector $\mathbf{1}_{y_i}$ with totally $C$ classes. Adversarial training (AT) can be formulated as a robust optimization problem (Madry et al., 2018):

$$\min_{\boldsymbol{\theta}} \sum_{i=1}^n \max_{\mathbf{x}_i' \in \mathcal{S}(\mathbf{x}_i)} \mathcal{L}(f_{\boldsymbol{\theta}}(\mathbf{x}_i'), y_i), \tag{1}$$

where $f_{\boldsymbol{\theta}}$ is a DNN classifier with parameters $\boldsymbol{\theta}$ that predicts probabilities over all classes, $\mathcal{L}$ is the classification loss (i.e., the cross-entropy loss as $\mathcal{L}(f_{\boldsymbol{\theta}}(\mathbf{x}), y) = -\mathbf{1}_y^\top \log f_{\boldsymbol{\theta}}(\mathbf{x})$), and $\mathcal{S}(\mathbf{x}) = \{\mathbf{x}' : \|\mathbf{x}' - \mathbf{x}\|_p \leq \epsilon\}$ is an adversarial region centered at $\mathbf{x}$ with radius $\epsilon > 0$ under the $\ell_p$-norm threat models (e.g., $\ell_2$ and $\ell_\infty$ norms that we consider). The robust optimization problem (1) is solved by using adversarial attacks to approximate the inner maximization and updating the model parameters

$\boldsymbol{\theta}$ via gradient descent. A typical method uses projected gradient descent (PGD) (Madry et al., 2018) for the inner problem, which starts at a randomly initialized point in $\mathcal{S}(\mathbf{x}_i)$ and iteratively updates the adversarial example under the $\ell_\infty$-norm threat model by

$$\mathbf{x}'_i = \Pi_{\mathcal{S}(\mathbf{x}_i)}\big(\mathbf{x}'_i + \alpha \cdot \operatorname{sign}(\nabla_{\mathbf{x}}\mathcal{L}(f_{\boldsymbol{\theta}}(\mathbf{x}'_i), y_i))\big), \tag{2}$$

where $\Pi(\cdot)$ is the projection operator and $\alpha$ is the step size.

Besides PGD-AT, another typical AT method is TRADES (Zhang et al., 2019b), which balances the trade-off between robustness and natural accuracy by minimizing a different adversarial loss

$$\min_{\boldsymbol{\theta}} \sum_{i=1}^{n} \left\{ \mathcal{L}(f_{\boldsymbol{\theta}}(\mathbf{x}_i), y_i) + \beta \cdot \max_{\mathbf{x}'_i \in \mathcal{S}(\mathbf{x}_i)} \mathcal{D}(f_{\boldsymbol{\theta}}(\mathbf{x}_i) \| f_{\boldsymbol{\theta}}(\mathbf{x}'_i)) \right\}, \tag{3}$$

where $\mathcal{L}$ is the clean cross-entropy loss on the natural example, $\mathcal{D}$ is the Kullback–Leibler divergence, and $\beta$ is a balancing parameter. The inner maximization of TRADES is also solved by PGD.

Recent progress of AT includes designing new adversarial losses (Mao et al., 2019; Qin et al., 2019; Pang et al., 2020; Wang et al., 2020; Dong et al., 2020a) and network architecture (Xie et al., 2019), training acceleration (Shafahi et al., 2019; Zhang et al., 2019a; Wong et al., 2020), and exploiting more training data (Hendrycks et al., 2019; Alayrac et al., 2019; Carmon et al., 2019; Zhai et al., 2019). Recent works highlight the training tricks in AT (Gowal et al., 2020; Pang et al., 2021).

## 2.2 RELATED WORK ON DNN MEMORIZATION

It has been observed that DNNs can easily memorize training data with random labels (Zhang et al., 2017), which requires "rethinking" of conventional techniques (e.g., VC dimension) to explain generalization. Arpit et al. (2017) identify qualitative differences between learning on true and random labels. Further works attempt to examine what and why DNNs memorize (Feldman, 2020; Feldman & Zhang, 2020; Maennel et al., 2020). Motivated by the memorization phenomenon in deep learning, convergence of training has been analyzed in the over-parameterized setting (Allen-Zhu et al., 2019; Du et al., 2019; Zou et al., 2020), while generalization has been studied with numerous theoretical and empirical complexity measures (Neyshabur et al., 2015; 2017; Bartlett et al., 2017; Novak et al., 2018; Arora et al., 2018; Cao & Gu, 2019; Jiang et al., 2020; Chen et al., 2020).

In contrast, the memorization behavior in AT has been less explored. The previous works demonstrate that DNNs can fit training data against an adversary (Madry et al., 2018; Schmidt et al., 2018; Rice et al., 2020), e.g., achieving nearly 100% robust training accuracy against a PGD adversary, but this behavior is not explored when trained on random labels. This paper is dedicated to investigating the memorization in AT under the extreme condition with random labels, while drawing connections to capacity, convergence, generalization, and robust overfitting, with the overarching goal of better understanding the AT working mechanism.

## 3 MEMORIZATION IN AT AND IMPLICATIONS

In this section, we first explore the memorization behavior in AT through an empirical study. Our analysis raises new questions about the convergence and generalization of AT, many of which cannot be answered by existing works. Thereafter, we provide further analytical studies on the convergence and generalization of AT by considering models trained on random labels particularly.

### 3.1 AT WITH RANDOM LABELS

We explore the memorization behavior of PGD-AT (Madry et al., 2018) and TRADES (Zhang et al., 2019b) as two studying cases. The experiments are conducted on CIFAR-10 (Krizhevsky & Hinton, 2009) with a Wide ResNet model (Zagoruyko & Komodakis, 2016) of depth 28 and widen factor 10 (WRN-28-10). Similar to Zhang et al. (2017), we train a network on the original dataset with true labels and on a copy of the dataset in which the true labels are corrupted by random ones. For training and robustness evaluation, a 10-step $\ell_\infty$ PGD adversary with $\epsilon = 8/255$ and $\alpha = 2/255$ is adopted. For TRADES, the PGD adversary maximizes the KL divergence during training, while maximizes the cross-entropy loss for robustness evaluation, as common practice (Zhang et al., 2019b). We set $\beta = 6.0$. In the sequel, we denote accuracy of a classifier against the 10-step PGD adversary as "**robust accuracy**", and accuracy on natural examples as "**natural accuracy**".

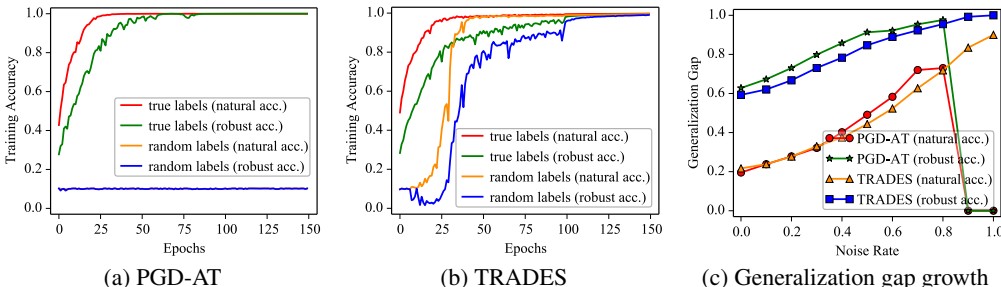

Figure 2: (a) and (b) show the natural and robust training accuracies of PGD-AT and TRADES, respectively, when trained on true or random labels. (c) shows the generalization gap under varying levels of label noise.

Fig. 2(a) and Fig. 2(b) show the learning curves of PGD-AT and TRADES without explicit regularizations. Both methods achieve almost $100\%$ natural and robust training accuracies when trained on true labels. When the labels are random, we observe the totally different behaviors between PGD-AT and TRADES—PGD-AT fails to converge while TRADES still reaches nearly $100\%$ training accuracies. This phenomenon is somewhat striking because PGD-AT and TRADES perform similarly on true labels (Rice et al., 2020). We find that the different memorization behaviors between PGD-AT and TRADES when trained on random labels can commonly be observed across a variety of datasets, model architectures, and threat models (shown in Appendix A.1), indicating that it is a general phenomenon of memorization in the two AT methods. Therefore, our finding is:

*DNNs have sufficient capacity to memorize adversarial examples of training data with completely random labels, but the convergence depends on the AT algorithms.*

**Partially corrupted labels.** We then inspect the behavior of AT under varying levels of label noise from $0\%$ (true labels) to $100\%$ (completely random labels). The generalization gap (i.e., difference between training and test accuracies) presented in Fig. 2(c) grows steadily as we increase the noise rate before the network fails to converge. The learning curves are provided in Appendix A.1.

**Explicit regularizations.** We study the role of common regularizers in AT memorization, including data augmentation, weight decay, and dropout (Srivastava et al., 2014). We train TRADES on true and random labels with several combinations of regularizers. We observe the explicit regularizers do not significantly affect the model's ability to memorize adversarial examples, similar to the finding in ST (Zhang et al., 2017; Arpit et al., 2017). The detailed results are provided in Appendix A.1.

### 3.2 Convergence analysis of AT with random labels

Since we have observed a counter-intuitive fact that PGD-AT and TRADES exhibit different convergence properties with random labels, it is necessary to perform a convergence analysis to understand this phenomenon. Note that our finding can hardly be explained by previous works (Gao et al., 2019; Wang et al., 2019; Zhang et al., 2020).

We first study the effects of different training settings on PGD-AT with random labels. We conduct experiments to analyze each training factor individually, including network architecture, attack steps, optimizer, and perturbation budget. We find that tuning the training settings cannot make PGD-AT converge with random labels (Appendix A.2 details the results). Based on the analysis, we think that the convergence issue of PGD-AT could be a result of the adversarial loss function in Eq. (1) rather than other training configurations. Specifically, TRADES in Eq. (3) minimizes a clean cross-entropy (CE) loss on natural examples, making DNNs memorize natural examples with random labels before fitting adversarial examples. As seen in Fig. 2(b), at the very early stage of TRADES training (the first 25 epochs), the natural accuracy starts to increase while the robust accuracy does not. However, PGD-AT in Eq. (1) directly minimizes the CE loss on adversarial samples with random labels, which can introduce unstable gradients with large variance, making it fail to converge. To corroborate the above argument, we analyze the gradient magnitude and stability below.

**Gradient magnitude.** First, we calculate the average gradient norm of the adversarial loss in Eq. (1) w.r.t. model parameters over each training sample for PGD-AT, and similarly calculate the average gradient norm of the clean CE loss (the first term) and the KL loss (the second term) in Eq. (3) w.r.t. parameters for TRADES to analyze their effects, respectively. We present the gradient norm along with training in Fig. 3(a). We can see that at the initial training epochs, the gradient norm of the KL loss in TRADES is much smaller than that of the CE loss, which indicates that the CE

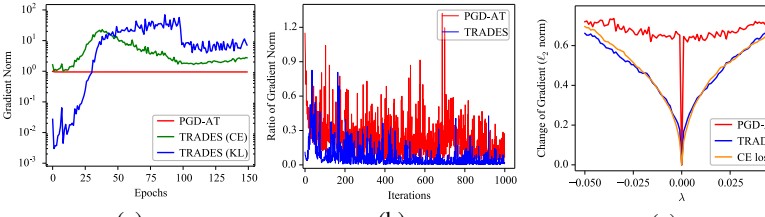

Figure 3: (a): Gradient norm of PGD-AT and TRADES along the training process. (b): The ratio of the gradient norm of PGD-AT and TRADES during the first 1000 training iterations.

Figure 4: (a): The $\ell_2$ distance between the gradients at $\boldsymbol{\theta}$ and $\boldsymbol{\theta} + \lambda\mathbf{d}$ of different losses, where $\boldsymbol{\theta}$ are initialized, $\lambda \in [-0.05, 0.05]$. (b): The cosine similarity between the gradients in each two successive epochs.

loss dominates TRADES training initially. With the training progressing, the KL loss has a larger gradient norm, making the network memorize adversarial examples. However, it is still unclear why PGD-AT does not rely on a similar learning tactic for convergence. To make a direct comparison with TRADES, we rewrite the adversarial loss of PGD-AT in Eq. (1) as

$$\max_{\mathbf{x}'_i \in \mathcal{S}(\mathbf{x}_i)} \mathcal{L}(f_{\boldsymbol{\theta}}(\mathbf{x}'_i), y_i) = \mathcal{L}(f_{\boldsymbol{\theta}}(\mathbf{x}_i), y_i) + \mathcal{R}(\mathbf{x}_i, y_i, \boldsymbol{\theta}),$$

where $\mathcal{R}(\mathbf{x}_i, y_i, \boldsymbol{\theta})$ denotes the difference between the CE loss on adversarial example $\mathbf{x}'_i$ and that on natural example $\mathbf{x}_i$. Hence we can separately calculate the gradient norm of $\mathcal{L}(f_{\boldsymbol{\theta}}(\mathbf{x}_i), y_i)$ and $\mathcal{R}(\mathbf{x}_i, y_i, \boldsymbol{\theta})$ w.r.t. parameters $\boldsymbol{\theta}$ to find out the effect of $\mathcal{R}(\mathbf{x}_i, y_i, \boldsymbol{\theta})$ on training. Specifically, we measure the relative gradient magnitude, i.e., in PGD-AT we calculate the ratio of the gradient norm $\frac{\|\nabla_{\boldsymbol{\theta}}\mathcal{R}(\mathbf{x}_i, y_i, \boldsymbol{\theta})\|_2}{\|\nabla_{\boldsymbol{\theta}}\mathcal{L}(f_{\boldsymbol{\theta}}(\mathbf{x}_i), y_i)\|_2}$; while in TRADES, we similarly calculate the ratio of the gradient norm of the KL loss to that of the CE loss. Fig. 3(b) illustrates the ratio of PGD-AT and TRADES during the first 1000 training iterations. The ratio of PGD-AT is consistently higher than that of TRADES, meaning that $\mathcal{R}(\mathbf{x}_i, y_i, \boldsymbol{\theta})$ has a non-negligible impact on training.

**Gradient stability.** Then, we analyze the gradient stability to explain why PGD-AT cannot converge. We denote the adversarial loss of PGD-AT as $\mathcal{J}(\mathbf{x}, y, \boldsymbol{\theta}) = \max_{\mathbf{x}' \in \mathcal{S}(\mathbf{x})} \mathcal{L}(f_{\boldsymbol{\theta}}(\mathbf{x}'), y)$ with the subscript $i$ omitted for notation simplicity. We have a theorem on gradient stability.

**Theorem 1.** *Suppose the gradient of the clean cross-entropy loss is locally Lipschitz continuous as*

$$\|\nabla_{\boldsymbol{\theta}}\mathcal{L}(f_{\boldsymbol{\theta}}(\mathbf{x}'), y) - \nabla_{\boldsymbol{\theta}}\mathcal{L}(f_{\boldsymbol{\theta}}(\mathbf{x}), y)\|_2 \leq K\|\mathbf{x}' - \mathbf{x}\|_p, \tag{4}$$

*for any $\mathbf{x} \in \mathbb{R}^d$, $\mathbf{x}' \in \mathcal{S}(\mathbf{x})$, and any $\boldsymbol{\theta}$, where $K$ is the Lipschitz constant. Then we have*

$$\|\nabla_{\boldsymbol{\theta}}\mathcal{J}(\mathbf{x}, y, \boldsymbol{\theta}_1) - \nabla_{\boldsymbol{\theta}}\mathcal{J}(\mathbf{x}, y, \boldsymbol{\theta}_2)\|_2 \leq \|\nabla_{\boldsymbol{\theta}}\mathcal{L}(f_{\boldsymbol{\theta}_1}(\mathbf{x}), y) - \nabla_{\boldsymbol{\theta}}\mathcal{L}(f_{\boldsymbol{\theta}_2}(\mathbf{x}), y)\|_2 + 2\epsilon K. \tag{5}$$

We provide the proof in Appendix B, where we show the upper bound in Eq. (5) is tight. Theorem 1 indicates that the gradient of the adversarial loss $\mathcal{J}(\mathbf{x}, y, \boldsymbol{\theta})$ of PGD-AT will change more dramatically than that of the clean CE loss $\mathcal{L}(f_{\boldsymbol{\theta}}(\mathbf{x}), y)$. When $\boldsymbol{\theta}_1$ and $\boldsymbol{\theta}_2$ are close, the difference between the gradients of $\mathcal{L}$ at $\boldsymbol{\theta}_1$ and $\boldsymbol{\theta}_2$ is close to $0$ due to the semi-smoothness of over-parameterized DNNs (Allen-Zhu et al., 2019), but that of $\mathcal{J}$ is relatively large due to $2\epsilon K$ in Eq. (5). To validate this, we visualize the change of gradient when moving the parameters $\boldsymbol{\theta}$ along a random direction $\mathbf{d}$ with magnitude $\lambda$. In particular, we set $\boldsymbol{\theta}$ as initialization, $\mathbf{d}$ is sampled from a Gaussian distribution and normalized filter-wise (Li et al., 2018). For PGD-AT and TRADES, we craft adversarial examples on-the-fly for the model with $\boldsymbol{\theta} + \lambda\mathbf{d}$ and measure the change of gradient by the $\ell_2$ distance to gradient at $\boldsymbol{\theta}$ averaged over all data samples. The curves on gradient change of PGD-AT, TRADES, and the clean CE loss are shown in Fig. 4(a). In a small neighborhood of $\boldsymbol{\theta}$ (i.e., small $\lambda$), the gradient of PGD-AT changes abruptly while the gradients of TRADES and the clean CE loss are more continuous. The gradient instability leads to a lower cosine similarity between the gradient directions w.r.t. the same data in each two successive training epochs of PGD-AT, as illustrated in Fig. 4(b). Therefore, the training of PGD-AT would be rather unstable that the gradient exhibits large variance, making it fail to converge.

**Clean CE loss helps PGD-AT converge.** To further verify our argument, we add the clean CE loss into the PGD-AT objective to resemble the learning of TRADES with random labels, as

$$\min_{\boldsymbol{\theta}} \sum_{i=1}^{n} \left\{ (1 - \gamma) \cdot \mathcal{L}(f_{\boldsymbol{\theta}}(\mathbf{x}_i), y_i) + \gamma \cdot \max_{\mathbf{x}'_i \in \mathcal{S}(\mathbf{x}_i)} \mathcal{L}(f_{\boldsymbol{\theta}}(\mathbf{x}'_i), y_i) \right\}, \tag{6}$$

Figure 5: AT by Eq. (6)

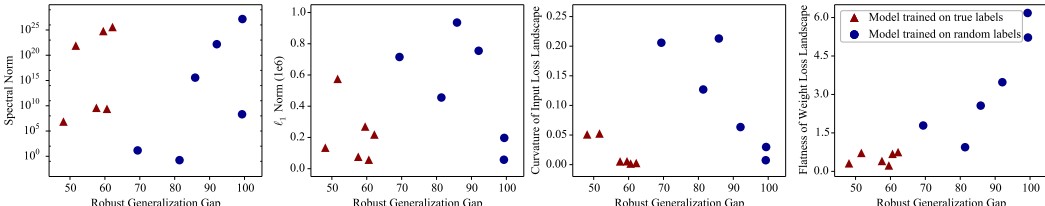

Figure 6: The results on four complexity measures of the adversarially trained models w.r.t. robust generalization gap. The training settings of these models are provided in Appendix A.3.

where $\gamma$ is gradually increased from 0 to 1. By using Eq. (6), the gradient would be stabler at the initial stage and training on random labels can successfully converge, as shown Fig. 5.

**In summary**, our convergence analysis identifies the gradient instability issue of PGD-AT, provides new insights on the differences between PGD-AT and TRADES, and partially explain the failures of AT under other realistic settings beyond the scope of this section as detailed in Appendix A.2.

### 3.3 GENERALIZATION ANALYSIS OF AT WITH RANDOM LABELS

As our study demonstrates DNNs' ability to memorize adversarial examples with random labels, we raise the question of whether DNNs rely on a similar memorization tactic on true labels and how to explain/ensure robust generalization. Although many efforts have been devoted to studying robust generalization of AT theoretically or empirically (Yin et al., 2018; Schmidt et al., 2018; Bubeck et al., 2019; Tu et al., 2019; Wu et al., 2020), they do not take the models trained on random labels into consideration. As it is easy to show that the explicit regularizations are not the adequate explanation of generalization in ST (Zhang et al., 2017; Arpit et al., 2017) and AT (see Appendix A.3), people resort to complexity measures of a model to explain generalization (i.e., a lower complexity should imply a smaller generalization gap). Here we show how the recently proposed complexity measures fail to explain robust generalization when comparing models trained on true and random labels.

We consider several norm-based and sharpness/flatness-based measures. We denote the parameters of a network by $\boldsymbol{\theta} := \{W_i\}_{i=1}^m$. The norm-based measures include spectral norm $\frac{1}{\gamma_{\text{margin}}} \prod_{i=1}^m \|W_i\|_2$ and $\ell_1$ norm $\frac{1}{\gamma_{\text{margin}}} \sum_{i=1}^m \|W_i\|_1$ of model parameters, where $\gamma_{\text{margin}}$ is a margin on model output to make them scale-insensitive (Neyshabur et al., 2017). The spectral norm appears in the theoretical robust generalization bounds (Yin et al., 2018; Tu et al., 2019) and is related to the Lipschitz constant of neural networks (Cisse et al., 2017). The $\ell_1$ norm is adopted to reduce the robust generalization gap (Yin et al., 2018). The sharpness/flatness-based measures include the curvature of input loss landscape (Moosavi-Dezfooli et al., 2019) as the dominant eigenvalue of the Hessian matrix, as well as the flatness of weight loss landscape (Wu et al., 2020) related to the change of adversarial loss when moving the weights along a random direction. Fig. 6 plots the four complexity measures w.r.t. robust generalization gap of several models trained with various combinations of regularizations on true or random labels. The results show that the first three measures can hardly ensure robust generalization, that lower complexity does not necessarily imply smaller robust generalization gap, e.g., the models trained on random labels can even lead to lower complexity than those trained on true labels. Among them, the flatness of weight loss landscape (Wu et al., 2020) is more reliable.

**In summary**, the generalization analysis indicates that the previous approaches, especially various complexity measures, cannot adequately explain and ensure the robust generalization performance in AT. Our finding of robust generalization in AT is complementary to that of standard generalization in ST (Zhang et al., 2017; Neyshabur et al., 2017; Jiang et al., 2020). Accordingly, robust generalization of adversarially trained models remains an open problem for future research.

## 4 ROBUST OVERFITTING ANALYSIS

Rice et al. (2020) have identified robust overfitting as a dominant phenomenon in AT, i.e., shortly after the first learning rate decay, further training will continue to decrease the robust test accuracy. They further show that several remedies for overfitting, including explicit $\ell_1$ and $\ell_2$ regularizations, data augmentation, etc., cannot gain improvements upon early stopping. Although robust overfitting has been thoroughly investigated, there still lacks an explanation of why it occurs. In this section, we draw a connection between memorization and robust overfitting in AT by showing that robust overfitting is caused by excessive memorization of one-hot labels in the typical AT methods. Motivated by the analysis, we then propose an effective strategy to eliminate robust overfitting.

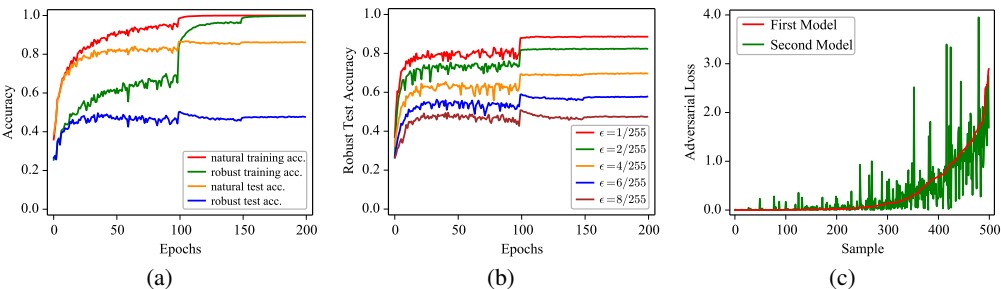

Figure 7: (a): The accuracy curves of PGD-AT with true labels to reproduce robust overfitting. (b): The robust test accuracy of PGD-AT under various perturbation budgets $\epsilon$. (c): The adversarial loss of two independently trained networks by PGD-AT on 500 samples sorted by the loss of the first model.

### 4.1 EXPLAINING ROBUST OVERFITTING

The typical AT approaches (e.g., PGD-AT, TRADES) commonly adopt one-hot labels as the targets for training, as introduced in Sec. 2.1. The one-hot labels could be inappropriate for some adversarial examples because it is difficult for a network to assign high-confident one-hot labels for all perturbed samples within the perturbation budget $\epsilon$ (Stutz et al., 2020; Cheng et al., 2020). Intuitively, some examples may naturally lie close to the decision boundary and should be assigned lower predictive confidence for the worst-case adversarial examples. It indicates that one-hot labels of some training data may be *noisy* in AT[1]. After a certain training epoch, the model memorizes these "hard" training examples with possibly noisy labels, leading to the reduction of test robustness, as shown in Fig. 7(a). Thus, we hypothesize the cause of robust overfitting lies in the *memorization of one-hot labels*.

Our hypothesis is well supported by two pieces of evidence. First, we find that when the perturbation budget $\epsilon$ is small, robust overfitting does not occur, as shown in Fig. 7(b). This observation implies that the one-hot labels are more appropriate as the targets for adversarial examples within a smaller neighborhood while become noisier under a larger perturbation budget and lead to overfitting. Second, we validate that the "hard" training examples with higher adversarial loss values are consistent across different models. We first train two independent networks (using the same architecture and different random seeds) by PGD-AT and calculate the adversarial loss for each training sample. We show the adversarial losses on 500 samples sorted by the loss of the first model in Fig. 7(c). It can be seen that the samples with lower adversarial losses of the first model also have relatively lower losses of the second one and vice versa. We further quantitatively measure the consistency of the adversarial losses of all training samples between the two models using the Kendall's rank coefficient (Kendall, 1938), which is $0.85$ in this case. A similar result can be observed for two different model architectures (see Appendix C.1). The results verify that the "hard" training examples with possibly noisy labels are intrinsic of a dataset, supporting our hypothesis on why robust overfitting occurs.

### 4.2 MITIGATING ROBUST OVERFITTING

Based on the above analysis, we resort to the methods that are less prone to overfit noisy labels for mitigating robust overfitting in AT. Although learning with noisy labels has been broadly studied in ST (Natarajan et al., 2013; Patrini et al., 2017; Jiang et al., 2018; Han et al., 2018; Zhang & Sabuncu, 2018), we find that most of these approaches are not suitable for AT. For example, a typical line of methods filter out noisy samples and train the models on the identified clean samples (Jiang et al., 2018; Han et al., 2018; Ren et al., 2018). However, they will neglect a portion of training data with noisy labels, which can lead to inferior results for AT due to the reduction of training data (Schmidt et al., 2018). Table 2 shows the results to validate this.

To address this problem, we propose to regularize the predictions of adversarial examples from being over-confident by integrating the **temporal ensembling (TE)** approach (Laine & Aila, 2017) into the AT frameworks. TE maintains an ensemble prediction of each data and penalizes the difference between the current prediction and the ensemble prediction, which is effective for semi-supervised learning and learning with noisy labels (Laine & Aila, 2017). We think that TE is suitable for AT since it enables to leverage all training samples and hinders the network from excessive memorization of one-hot labels with a regularization term. Specifically, we denote the ensemble prediction of

---

[1]Note that we argue the one-hot labels are noisy when used in AT, but do not argue the ground-truth labels of the dataset are noisy, which is different from a previous work (Sanyal et al., 2021).

Table 1: Test accuracy (%) of several methods on CIFAR-10, CIFAR-100, and SVHN under the $\ell_\infty$ norm with $\epsilon = 8/255$ based on the ResNet-18 architecture. We choose the best checkpoint according to the highest robust accuracy on the test set under PGD-10.

| Method | Natural Accuracy | | | PGD-10 | | | PGD-1000 | | | C&W-1000 | | | AutoAttack | | |
|---|---|---|---|---|---|---|---|---|---|---|---|---|---|---|---|
| | Best | Final | Diff | Best | Final | Diff | Best | Final | Diff | Best | Final | Diff | Best | Final | Diff |
| PGD-AT | **83.75** | 84.82 | -1.07 | 52.64 | 44.92 | 7.72 | 51.22 | 42.74 | 8.48 | 50.11 | 43.63 | 7.48 | 47.74 | 41.84 | 5.90 |
| PGD-AT+TE | 82.35 | 82.79 | **-0.44** | **55.79** | **54.83** | **0.96** | **54.65** | **53.30** | **1.35** | **52.30** | **51.73** | **0.57** | **50.59** | **49.62** | **0.97** |
| TRADES | 81.19 | 82.48 | -1.29 | 53.32 | 50.25 | 3.07 | 52.44 | 48.67 | 3.77 | 49.88 | 48.14 | 1.74 | 49.03 | 46.80 | 2.23 |
| TRADES+TE | **83.86** | **83.97** | **-0.11** | **55.15** | **54.42** | **0.73** | **53.74** | **53.03** | **0.71** | **50.77** | **50.63** | **0.14** | **49.77** | **49.20** | **0.57** |

(a) The evaluation results on **CIFAR-10**.

| Method | Natural Accuracy | | | PGD-10 | | | PGD-1000 | | | C&W-1000 | | | AutoAttack | | |
|---|---|---|---|---|---|---|---|---|---|---|---|---|---|---|---|
| | Best | Final | Diff | Best | Final | Diff | Best | Final | Diff | Best | Final | Diff | Best | Final | Diff |
| PGD-AT | **57.54** | 57.51 | 0.03 | 29.40 | 21.75 | 7.65 | 28.54 | 20.63 | 7.91 | 27.06 | 21.17 | 5.89 | 24.72 | 19.34 | 5.38 |
| PGD-AT+TE | 56.45 | 57.12 | -0.67 | **31.74** | **30.24** | **1.50** | **31.27** | **29.80** | **1.47** | **28.27** | **27.36** | **0.91** | **26.30** | **25.34** | **0.96** |
| TRADES | 57.98 | 56.32 | 1.66 | 29.93 | 27.70 | 2.23 | 29.51 | 26.93 | 2.58 | 25.46 | 24.42 | 1.04 | 24.61 | 23.40 | 1.21 |
| TRADES+TE | **59.35** | **58.72** | **0.63** | **31.09** | **30.12** | **0.97** | **30.54** | **29.45** | **1.09** | **26.61** | **25.94** | **0.67** | **25.27** | **24.55** | **0.72** |

(b) The evaluation results on **CIFAR-100**.

| Method | Natural Accuracy | | | PGD-10 | | | PGD-1000 | | | C&W-1000 | | | AutoAttack | | |
|---|---|---|---|---|---|---|---|---|---|---|---|---|---|---|---|
| | Best | Final | Diff | Best | Final | Diff | Best | Final | Diff | Best | Final | Diff | Best | Final | Diff |
| PGD-AT | 89.00 | 90.55 | -1.55 | 54.51 | 46.97 | 7.54 | 52.22 | 42.85 | 9.37 | 48.66 | 44.13 | 4.53 | 46.61 | 38.24 | 8.37 |
| PGD-AT+TE | **90.09** | **90.91** | **-0.82** | **59.74** | **59.05** | **0.69** | **57.71** | **56.46** | **1.25** | **54.55** | **53.94** | **0.61** | **51.44** | **50.61** | **0.83** |
| TRADES | **90.88** | **91.30** | **-0.42** | 59.50 | 57.04 | 2.46 | 52.78 | 50.17 | 2.61 | 52.76 | 50.53 | 2.23 | 40.36 | 38.88 | 1.48 |
| TRADES+TE | 89.01 | 88.52 | 0.49 | **59.81** | **58.49** | **1.32** | **58.24** | **56.66** | **1.58** | **54.00** | **53.24** | **0.76** | **51.45** | **50.16** | **1.29** |

(c) The evaluation results on **SVHN**.

a training sample $\mathbf{x}_i$ as $\mathbf{p}_i$, which is updated in each training epoch as $\mathbf{p}_i \leftarrow \eta \cdot \mathbf{p}_i + (1 - \eta) \cdot f_{\boldsymbol{\theta}}(\mathbf{x}_i)$, where $\eta$ is the momentum term. The training objective of PGD-AT with TE can be expressed as

$$\min_{\boldsymbol{\theta}} \sum_{i=1}^n \max_{\mathbf{x}_i' \in \mathcal{S}(\mathbf{x}_i)} \left\{ \mathcal{L}(f_{\boldsymbol{\theta}}(\mathbf{x}_i'), y_i) + w \cdot \|f_{\boldsymbol{\theta}}(\mathbf{x}_i') - \hat{\mathbf{p}}_i\|_2^2 \right\}, \tag{7}$$

where $\hat{\mathbf{p}}_i$ is the normalization of $\mathbf{p}_i$ as a probability vector and $w$ is a balancing weight. TE can be similarly integrated with TRADES with the same regularization term. The network would learn to fit relatively easy samples with one-hot labels in the initial training stage, as shown in Fig. 7(a). After the learning rate decays, the network can keep assigning low confidence for hard samples with the regularization term in Eq. (7) and avoid fitting one-hot labels. Therefore, the proposed algorithm enables to learn under label noise in AT and alleviates the robust overfitting problem.

## 5 EMPIRICAL EVALUATION ON MITIGATING ROBUST OVERFITTING

In this section, we provide the experimental results on CIFAR-10, CIFAR-100 (Krizhevsky & Hinton, 2009), and SVHN (Netzer et al., 2011) datasets to validate the effectiveness of our proposed method. Code is available at https://github.com/dongyp13/memorization-AT.

**Training details.** We adopt the common setting that the perturbation budget is $\epsilon = 8/255$ under the $\ell_\infty$ norm in most experiments. We consider PGD-AT and TRADES as two typical AT baselines and integrate the proposed TE approach into them, respectively. We use the ResNet-18 (He et al., 2016) model as the classifier in most experiments. In training, we use the 10-step PGD adversary with $\alpha = 2/255$. The models are trained via the SGD optimizer with momentum 0.9, weight decay 0.0005, and batch size 128. For CIFAR-10/100, we set the learning rate as 0.1 initially which is decayed by 0.1 at 100 and 150 epochs with totally 200 training epochs. For SVHN, the learning rate starts from 0.01 with a cosine annealing schedule for a total number of 80 training epochs. In our method, We set $\eta = 0.9$ and $w = 30$ along a Gaussian ramp-up curve (Laine & Aila, 2017).

**Evaluation results.** We adopt PGD-10, PGD-1000, C&W-1000 (Carlini & Wagner, 2017), and AutoAttack (Croce & Hein, 2020b) for evaluating adversarial robustness rigorously. AutoAttack is a strong attack to evaluate model robustness, which is composed of an ensemble of diverse attacks, including APGD-CE (Croce & Hein, 2020b), APGD-DLR (Croce & Hein, 2020b), FAB (Croce

& Hein, 2020a), and Square attack (Andriushchenko et al., 2020). To show the performance of robust overfitting, we report the test accuracy on the best checkpoint that achieves the highest robust test accuracy under PGD-10 and the final checkpoint, as well as the difference between these two checkpoints. The results of PGD-AT, TRADES, and the combinations of them with our proposed approach (denoted as PGD-AT**+TE** and TRADES**+TE**) on the CIFAR-10, CIFAR-100, and SVHN datasets are shown in Table 1.

We can observe that the differences between best and final test accuracies of our method are reduced to around 1%, while the accuracy gaps of PGD-AT and TRADES are much larger. It indicates that our method largely eliminates robust overfitting. Due to being less affected by robust overfitting, our method achieves higher robust accuracies than the baselines. We also show the learning curves of these methods in Fig. 8. We consistently demonstrate the effectiveness of our method on different network architectures (including WRN-34-10 and VGG-16) and threat models (including $\ell_2$ norm), which will be shown in Appendix C.2.

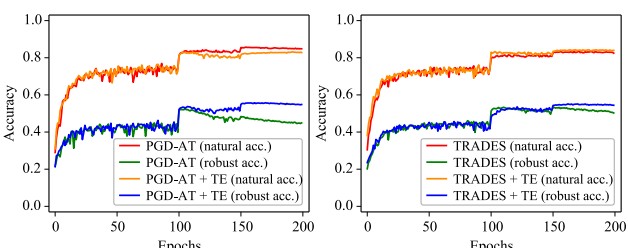

Figure 8: The natural and robust test accuracy curves (under PGD-10) of PGD-AT, TRADES, and their extensions by integrating the proposed TE approach. The models are trained on CIFAR-10 under the $\ell_\infty$ norm with $\epsilon = 8/255$ based on the ResNet-18 architecture.

**Discussion and comparison with related works.** Our method is kind of similar to the label smoothing (LS) technique, which is studied in AT (Pang et al., 2021). Recent works have also introduced the smoothness in training labels and model weights (Chen et al., 2021; Huang et al., 2020), which can alleviate robust overfitting to some extent. The significant difference between our work and them is that we provide a reasonable explanation for robust overfitting—*one-hot labels are noisy for AT*, while previous methods did not give such an explanation and could be viewed as solutions to our identified problem. To empirically compare with these methods, we conduct experiments on CIFAR-10 with the ResNet-18 network. Under the PGD-AT framework, we compare with the baseline PGD-AT, PGD-AT**+LS**, self-adaptive training (SAT) (Huang et al., 2020), and knowledge distillation with stochastic weight averaging (KD-SWA) (Chen et al., 2021). We also adopt the *Co-teaching* approach (Han et al., 2018) adapted to PGD-AT, which jointly trains two models using the filtered samples given by each other. The results under the adopted attacks are presented in Table 2. Although various techniques can alleviate robust overfitting, our method achieves better robustness than the others, validating its effectiveness. For Co-teaching, though robust overfitting is alleviated, the performance is worse than our proposed method due to the reduction of training data.

Table 2: Test accuracy (%) of the proposed method and other methods on CIFAR-10 under the $\ell_\infty$ norm with $\epsilon = 8/255$ based on the ResNet-18 architecture.

| Method | Natural Accuracy | | | PGD-10 | | | PGD-1000 | | | C&W-1000 | | | AutoAttack | | |
|---|---|---|---|---|---|---|---|---|---|---|---|---|---|---|---|
| | Best | Final | Diff | Best | Final | Diff | Best | Final | Diff | Best | Final | Diff | Best | Final | Diff |
| PGD-AT | 83.75 | 84.82 | -1.07 | 52.64 | 44.92 | 7.72 | 51.22 | 42.74 | 8.48 | 50.11 | 43.63 | 7.48 | 47.74 | 41.84 | 5.90 |
| PGD-AT**+LS** | 82.68 | 85.16 | -2.48 | 53.70 | 48.90 | 4.80 | 52.56 | 46.31 | 6.25 | 50.41 | 46.06 | 4.35 | 49.02 | 44.39 | 4.63 |
| SAT | 82.81 | 81.86 | 0.95 | 53.81 | 53.31 | **0.50** | 52.41 | 52.00 | **0.41** | 51.99 | 51.71 | **0.28** | 50.21 | 49.73 | **0.48** |
| KD-SWA | **84.84** | **85.26** | -0.42 | 54.89 | 53.80 | 1.09 | 53.31 | 52.45 | 0.86 | 51.48 | 50.91 | 0.57 | 50.42 | **49.83** | 0.59 |
| Co-teaching | 81.94 | 82.22 | **-0.28** | 51.27 | 50.52 | 0.75 | 50.15 | 49.12 | 1.03 | 50.85 | 49.86 | 0.99 | 49.60 | 48.49 | 1.11 |
| PGD-AT**+TE** | 82.35 | 82.79 | -0.44 | **55.79** | **54.83** | 0.96 | **54.65** | **53.30** | 1.35 | **52.30** | **51.73** | 0.57 | **50.59** | 49.62 | 0.97 |

## 6 CONCLUSION

In this paper, we demonstrate the capacity of DNNs to fit adversarial examples with random labels by exploring memorization in adversarial training, which also poses open questions on the convergence and generalization of adversarially trained models. We validate that some AT methods suffer from a gradient instability issue and robust generalization can hardly be explained by complexity measures. We further identify a significant drawback of memorization in AT related to the robust overfitting phenomenon—robust overfitting is caused by memorizing one-hot labels in adversarial training. We propose a new mitigation algorithm to address this issue, with the effectiveness validated extensively.

## ACKNOWLEDGEMENTS

This work was supported by the National Key Research and Development Program of China (2020AAA0106000, 2020AAA0104304, 2020AAA0106302), NSFC Projects (Nos. 61620106010, 62061136001, 61621136008, 62076147, U19B2034, U1811461, U19A2081), Beijing NSF Project (No. JQ19016), Beijing Academy of Artificial Intelligence (BAAI), Tsinghua-Alibaba Joint Research Program, Tsinghua Institute for Guo Qiang, Tsinghua-OPPO Joint Research Center for Future Terminal Technology.

## ETHICS STATEMENT

The existence of adversarial examples can pose severe security threats to machine learning and deep learning models when they are deployed to real-world applications. The vulnerability to adversarial examples could also lower the confidence of the public on machine learning techniques. Therefore, it is important to develop more robust models. As the most effective method for promoting model robustness, adversarial training (AT) has not been fully investigated. This paper aims to investigate the memorization effect of AT to facilitate a better understanding of its working mechanism. Some findings in this paper can be analyzed more deeply, including theoretical analysis of AT convergence, generalization, etc., which we leave to future work.

## REPRODUCIBILITY STATEMENT

Most of the experiments are easily reproducible. We provide the code for reproducing the results at https://github.com/dongyp13/memorization-AT.

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

## A    ADDITIONAL EXPERIMENTS ON MEMORIZATION IN AT

In this section, we provide additional experiments on the memorization behavior in AT. All of the experiments are conducted on NVIDIA 2080 Ti GPUs. The source code of this paper is submitted as the supplementary material, and will be released after the review process.

### A.1    MEMORIZATION OF PGD-AT AND TRADES

#### A.1.1    DIFFERENT TRAINING SETTINGS

We first demonstrate that the different memorization behaviors between PGD-AT and TRADES can be generally observed under various settings.

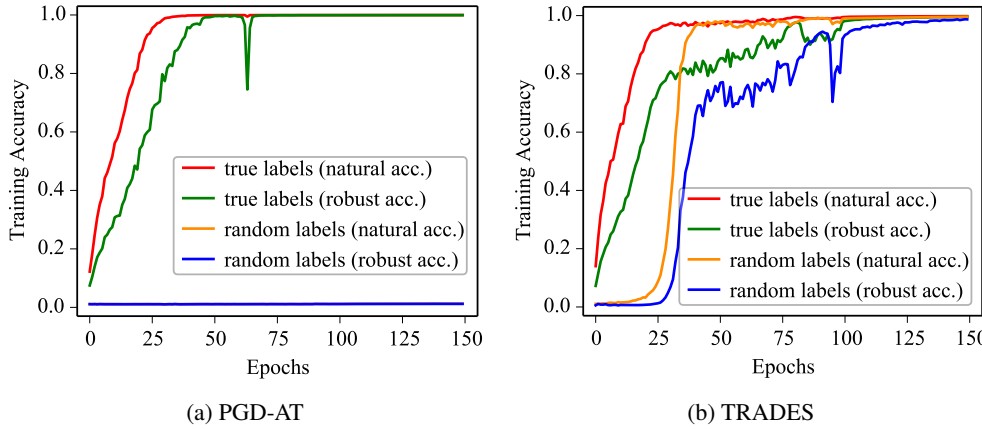

(a) PGD-AT                                    (b) TRADES

Figure A.1: The natural and robust training accuracies of PGD-AT and TRADES on CIFAR-100 when trained on true or random labels.

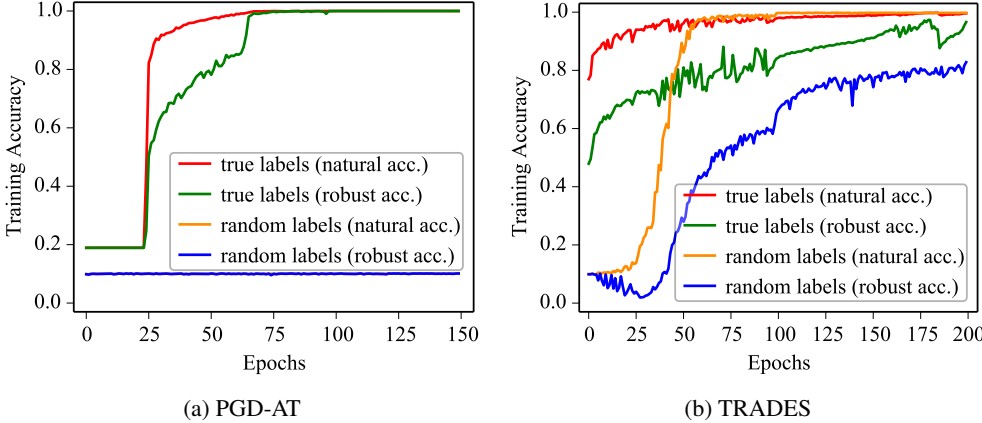

(a) PGD-AT                                    (b) TRADES

Figure A.2: The natural and robust training accuracies of PGD-AT and TRADES on SVHN when trained on true or random labels.

**Datasets.** Similar to Fig. 2, we show the accuracy curves of PGD-AT and TRADES when trained on true or random labels on CIFAR-100 (Krizhevsky & Hinton, 2009) in Fig. A.1 and on SVHN (Netzer et al., 2011) in Fig. A.2. We consistently observe that PGD-AT fails to converge with random labels, while TRADES can successfully converge, although it does not reach 100% accuracy on SVHN.

**Model architectures.** We then consider other network architectures, including the DenseNet-121 model (Huang et al., 2017) and the deep layer aggregation (DLA) model (Yu et al., 2018). The corresponding results are shown in Fig. A.3. The similar results can be observed, although it may take more training epochs to make TRADES converge with the smaller DenseNet-121 network.

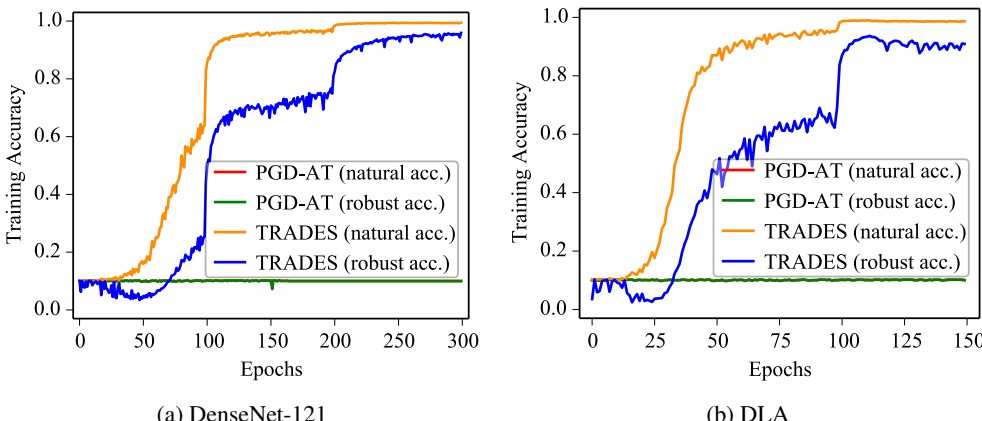

(a) DenseNet-121                                       (b) DLA

Figure A.3: The natural and robust training accuracies of PGD-AT and TRADES on CIFAR-10 with different architectures when trained on random labels.

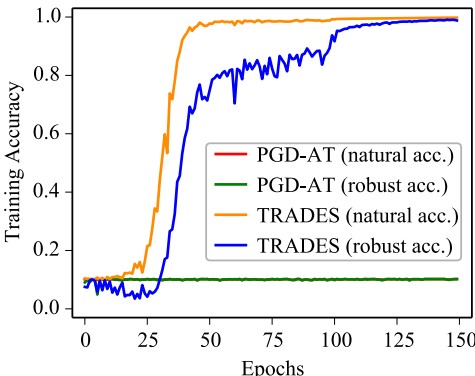

Figure A.4: The natural and robust training accuracies of PGD-AT and TRADES on CIFAR-10 under the $\ell_2$-norm threat model when trained on random labels.

**Threat models.** We further consider the $\ell_2$-norm threat model, in which we set $\epsilon = 1.0$ and $\alpha = 0.25$ in the 10-step PGD adversary. The learning curves of PGD-AT and TRADES are shown in Fig. A.4, which also exhibit similar results.

**Perturbation budget.** We study the memorization behavior in AT with different perturbation budgets $\epsilon$. In Fig. A.5, we show that when the perturbation budget is $\epsilon = 16/255$ ($\ell_\infty$ norm), TRADES trained on random labels can still converge. But when we set a larger budget (e.g., $\epsilon = 32/255$), both PGD-AT and TRADES cannot obtain near 100% robust training accuracy. We also find under this condition, even AT trained on true labels cannot get 100% robust training accuracy, indicating that the gradient instability issue discussed in Sec. 3.2 results in the convergence problem.

In summary, our empirical observation that PGD-AT and TRADES perform differently when trained on random labels is general across multiple datasets, network architectures, and threat models.

### A.1.2 LEARNING CURVES UNDER DIFFERENT NOISE RATES

We show the learning curves of PGD-AT and TRADES under varying levels of label noise in Fig. A.6 and Fig. A.7, respectively. In this experiment, we adopt the weight decay and data augmentation for regularizations. We can see that the network achieves maximum accuracy on the test set before fitting the noisy training set. Thus the model learns easy and simple patterns first before fitting the noise, similar to the finding in ST (Arpit et al., 2017). It can also be observed that under 80% noise rate, PGD-AT fails to converge. Note that when the noise rate is 0%, the network is trained on true labels, but the robust test accuracy also decreases after a certain epoch. This phenomenon is called robust overfitting (Rice et al., 2020), which is studied in Sec. 4.

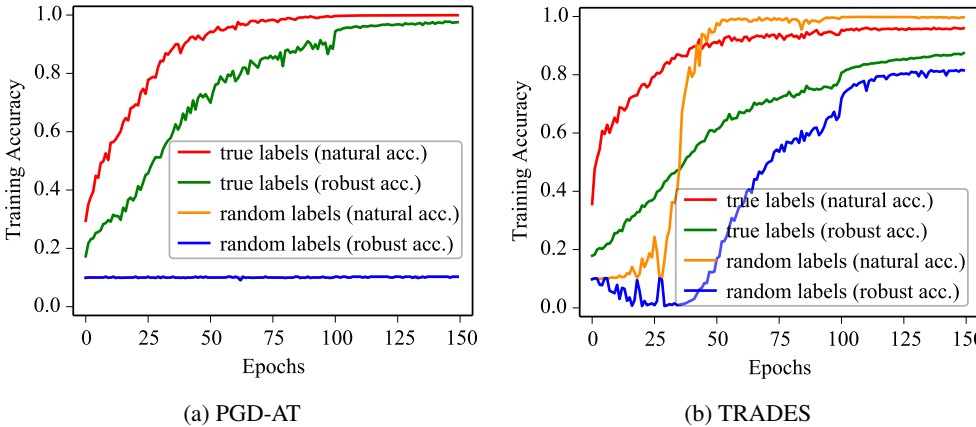

(a) PGD-AT               (b) TRADES

Figure A.5: The natural and robust training accuracies of PGD-AT and TRADES on CIFAR-10 with $\epsilon = 16/255$ when trained on true or random labels.

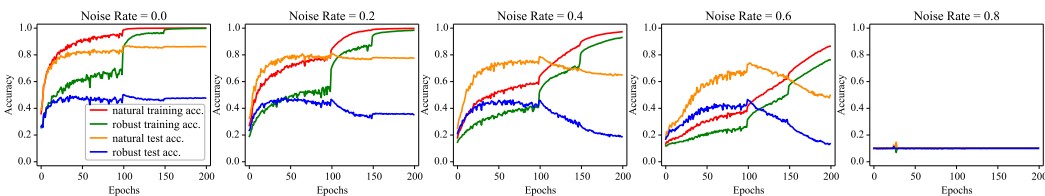

Figure A.6: Accuracy curves of PGD-AT under different noise rates on CIFAR-10.

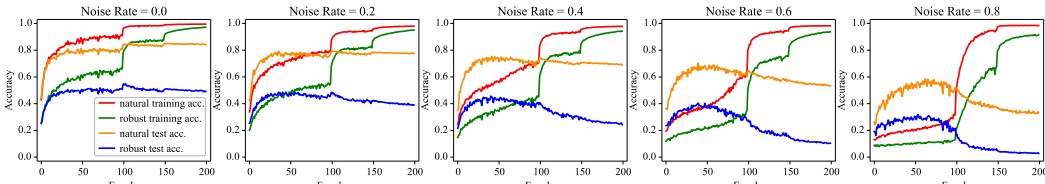

Figure A.7: Accuracy curves of TRADES under different noise rates on CIFAR-10.

### A.1.3 EXPLICIT REGULARIZATIONS

We consider three common regularizers, including data augmentation, weight decay, and dropout (Srivastava et al., 2014). We train the models based on TRADES on true and random labels with several combinations of explicit regularizers. As shown in Table A.1, the explicit regularizations do not significantly affect the model's ability to memorize adversarial examples with random labels.

### A.2 MORE RESULTS ON THE CONVERGENCE OF AT

### A.2.1 TRAINING CONFIGURATIONS

We study different training configurations on PGD-AT with random labels. We consider various factors as follows. These experiments are conducted on CIFAR-10.

- **Model capacity.** Recent work suggests that model size is a critical factor to obtain better robustness (Madry et al., 2018; Xie & Yuille, 2020). A possible reason why PGD-AT fails to converge with random labels may also be the insufficient model capacity. Therefore, we try to use larger models, including WRN-34-20 (which is used in Rice et al. (2020)) and WRN-70-16 (which is used in Gowal et al. (2020)). However, using larger models under this setting cannot solve the convergence problem.

Table A.1: The training accuracy, test accuracy, and generalization gap (%) of TRADES when trained on true or random labels, with and without explicit regularizations, including data augmentation (random crop and flip), weight decay (0.0002), and dropout (0.2).

| Labels | Data Augmentation | Weight Decay | Dropout | Training Accuracy | | Test Accuracy | | Generalization Gap | |
|--------|-------------------|--------------|---------|-------------------|--------|---------------|--------|--------------------|--------|
| | | | | Natural | Robust | Natural | Robust | Natural | Robust |
| true | ✗ | ✗ | ✗ | 99.73 | 99.65 | 77.53 | 37.47 | 22.20 | 62.18 |
| true | ✓ | ✗ | ✗ | 99.57 | 97.03 | 82.91 | 45.37 | 16.93 | 51.66 |
| true | ✗ | ✓ | ✗ | 99.59 | 99.53 | 77.31 | 38.94 | 22.28 | 60.59 |
| true | ✗ | ✗ | ✓ | 99.65 | 99.40 | 79.96 | 39.86 | 19.69 | 59.54 |
| true | ✓ | ✓ | ✗ | 99.50 | 97.28 | 84.26 | 49.16 | 15.24 | 48.12 |
| true | ✗ | ✓ | ✓ | 99.41 | 99.20 | 80.28 | 41.64 | 19.13 | 57.56 |
| random | ✗ | ✗ | ✗ | 99.80 | 99.55 | 9.79 | 0.15 | 90.01 | 99.40 |
| random | ✓ | ✗ | ✗ | 99.36 | 86.10 | 9.71 | 0.24 | 89.65 | 85.86 |
| random | ✗ | ✓ | ✗ | 99.84 | 99.53 | 10.13 | 0.23 | 89.71 | 99.30 |
| random | ✗ | ✗ | ✓ | 99.15 | 92.23 | 9.04 | 0.17 | 90.11 | 92.06 |
| random | ✓ | ✓ | ✗ | 99.25 | 69.62 | 9.67 | 0.24 | 89.58 | 69.38 |
| random | ✗ | ✓ | ✓ | 99.38 | 81.57 | 9.54 | 0.19 | 89.84 | 81.38 |

- **Attack steps.** We adopt the weaker FGSM adversary (Goodfellow et al., 2015) for training. We also adopt the random initialization trick as argued in Wong et al. (2020) and adjust the step size as $\alpha = 10/255$, yielding the fast adversarial training method (Wong et al., 2020). However, fast AT still cannot converge.

- **Optimizer.** We try to use various optimizers, including the SGD momentum optimizer, the Adam optimizer (Kingma & Ba, 2015), and the nesterov optimizer (Nesterov, 1983); different learning rate schedules, including the piecewise decay and cosine schedules, and different learning rates (0.1 and 0.01), but none of these attempts make PGD-AT converge.

- **Perturbation budget.** The perturbation budget $\epsilon$ is an important factor to affect the convergence of PGD-AT. When $\epsilon$ approaches 0, PGD-AT would degenerate into standard training, which can easily converge (Zhang et al., 2017). Hence we try different values of $\epsilon$, and find that PGD-AT can converge with a smaller $\epsilon$ (e.g., $\epsilon = 1/255$) but cannot converge when $\epsilon \geq 2/255$.

### A.2.2 GRADIENT STABILITY UNDER COSINE SIMILARITY

In Fig. 4(a), we show the gradient change of PGD-AT, TRADES, and the clean CE loss under the $\ell_2$ distance. We further show the cosine similarity between the gradients at $\boldsymbol{\theta}$ and $\boldsymbol{\theta} + \lambda \mathbf{d}$ in Fig. A.8. The cosine similarity is also averaged over all data samples. The results based on cosine similarity are consistent with the results based on the $\ell_2$ distance, showing that the gradient of PGD-AT changes more abruptly.

### A.2.3 THE FAILURES OF AT UNDER REALISTIC SETTINGS

We find that some AT methods (e.g., PGD-AT) suffer from a gradient instability issue, which results in the convergence problem when trained on random labels. Under other realistic setting, our analysis may also be valuable.

First, PGD-AT fails to converge under 80% noise rate. We think that the unstable gradients can overwhelm the useful gradients given by clean examples. To prove it, we train the models under 80% uniform label noise, by either PGD-AT or standard training (ST) on natural examples. We then select 100 training images with wrong labels and another 100 training images with true labels for evaluation. Similarly, we calculate the gradient norm of the cross-entropy loss w.r.t. model parameters of each method. We show the results in Fig. A.9. For AT, the gradient norm of clean examples is larger than that of noisy examples at beginning, which makes the model learn to classify. However, for PGD-AT, the gradient norm of clean examples is almost the same as that of noisy examples (the two curves overlap together). And the unstable gradients provided by noisy examples would overwhelm the useful gradients given by clean examples, making the network fail to converge.

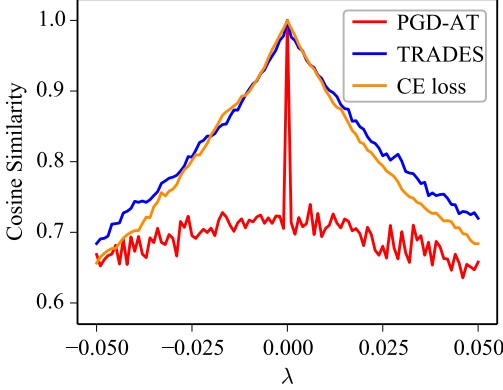 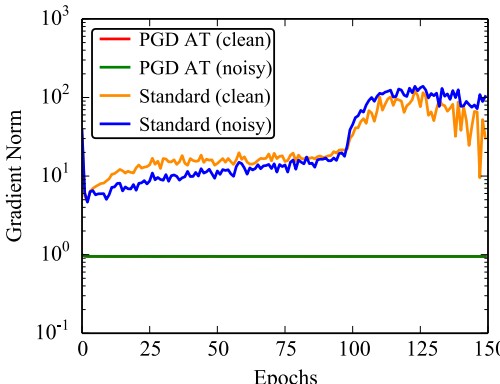

Figure A.8: The cosine similarity between the gradients at $\theta$ and $\theta + \lambda\mathbf{d}$ of different losses, where $\theta$ are initialized, $\lambda \in [-0.05, 0.05]$.

Figure A.9: The gradient norm of PGD-AT and ST given clean examples or noisy examples, when trained on 80% uniform label noise.

Second, when the perturbation budget is large (e.g., $\epsilon = 64/255$), PGD-AT cannot converge with true labels, while TRADES can achieve about 50% training accuracies. This can also be explained by our convergence analysis that the gradient is very unstable in PGD-AT with a larger perturbation budget, making it fail to converge.

### A.3 MORE DISCUSSIONS ON THE GENERALIZATION OF AT

As shown in Table A.1, when trained on true labels, although the regularizers can help to reduce the generalization gap, the model without any regularization can still generalize non-trivially. The three explicit regularizations do not significantly affect the model's ability to memorize adversarial examples with random labels. In consequence, the explicit regularizers are not the adequate explanation of generalization. By inspecting the learning dynamics of AT under different noise rates in Fig. A.6 and Fig. A.7, the network achieves maximum accuracy on the test set before fitting the noisy training set, meaning that the model learns simple patterns (i.e., clean data) before memorizing the hard examples with wrong labels, similar to the observation in standard training (Arpit et al., 2017). The results suggest that optimization by itself serves as an implicit regularizer to find a model with good generalization performance.

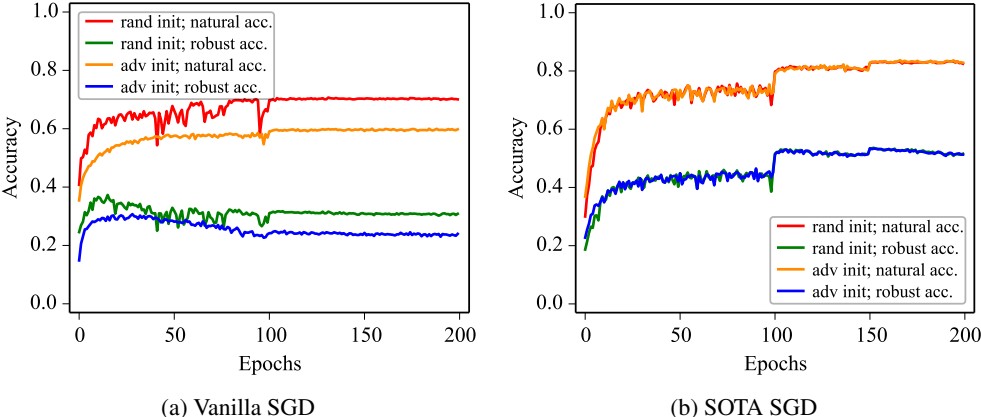

(a) Vanilla SGD

(b) SOTA SGD

Figure A.10: The natural and robust testing accuracies of TRADES on CIFAR-10 with different initialization strategies and training methods.

A recent work (Liu et al., 2020b) points out that in standard training, pre-training on random labels can lead to substantial performance degeneration of subsequent SGD training on true labels, while adding regularizations can overcome the bad initialization caused by pre-training with random labels. In this paper, we further investigate whether this finding can generalize to adversarial training.

As PGD-AT cannot converge with random labels, we adopt TRADES to conduct experiments. Following Liu et al. (2020b), we consider two initialization strategies — random initialization and adversarial initialization generated by training on random labeling of the training data. We also consider two training methods — vanilla SGD training and SOTA SGD training with data augmentation (random crops and flips), weight decay, and momentum. The results are shown in Fig. A.10. It can be seen that with vanilla SGD, the adversarial initialization can lead to worse performance than the random initialization. But with the regularization techniques, the models with different initializations converge to nearly the same test accuracy. The results are consistent with the findings in Liu et al. (2020b).

## B    PROOF OF THEOREM 1

*Proof.* Recall that $\mathcal{J}(\mathbf{x}, y, \boldsymbol{\theta}) = \max_{\mathbf{x}' \in \mathcal{S}(\mathbf{x})} \mathcal{L}(f_{\boldsymbol{\theta}}(\mathbf{x}'), y)$ is the adversarial loss of PGD-AT. First, we have

$$
\begin{aligned}
& \|\nabla_{\boldsymbol{\theta}} \mathcal{J}(\mathbf{x}, y, \boldsymbol{\theta}_1) - \nabla_{\boldsymbol{\theta}} \mathcal{J}(\mathbf{x}, y, \boldsymbol{\theta}_2)\|_2 \\
= & \|\nabla_{\boldsymbol{\theta}} \mathcal{J}(\mathbf{x}, y, \boldsymbol{\theta}_1) - \nabla_{\boldsymbol{\theta}} \mathcal{L}(f_{\boldsymbol{\theta}_1}(\mathbf{x}), y) - \\
& \nabla_{\boldsymbol{\theta}} \mathcal{J}(\mathbf{x}, y, \boldsymbol{\theta}_2) + \nabla_{\boldsymbol{\theta}} \mathcal{L}(f_{\boldsymbol{\theta}_2}(\mathbf{x}), y) + \\
& \nabla_{\boldsymbol{\theta}} \mathcal{L}(f_{\boldsymbol{\theta}_1}(\mathbf{x}), y) - \nabla_{\boldsymbol{\theta}} \mathcal{L}(f_{\boldsymbol{\theta}_2}(\mathbf{x}), y)\|_2 \\
\leq & \|\nabla_{\boldsymbol{\theta}} \mathcal{J}(\mathbf{x}, y, \boldsymbol{\theta}_1) - \nabla_{\boldsymbol{\theta}} \mathcal{L}(f_{\boldsymbol{\theta}_1}(\mathbf{x}), y)\|_2 + \\
& \|\nabla_{\boldsymbol{\theta}} \mathcal{J}(\mathbf{x}, y, \boldsymbol{\theta}_2) - \nabla_{\boldsymbol{\theta}} \mathcal{L}(f_{\boldsymbol{\theta}_2}(\mathbf{x}), y)\|_2 + \\
& \|\nabla_{\boldsymbol{\theta}} \mathcal{L}(f_{\boldsymbol{\theta}_1}(\mathbf{x}), y) - \nabla_{\boldsymbol{\theta}} \mathcal{L}(f_{\boldsymbol{\theta}_2}(\mathbf{x}), y)\|_2.
\end{aligned}
\tag{B.1}
$$

From the assumption, for any $\mathbf{x} \in \mathbb{R}^d$ and $\mathbf{x}' \in \mathcal{S}(\mathbf{x})$, we have

$$
\|\nabla_{\boldsymbol{\theta}} \mathcal{L}(f_{\boldsymbol{\theta}}(\mathbf{x}'), y) - \nabla_{\boldsymbol{\theta}} \mathcal{L}(f_{\boldsymbol{\theta}}(\mathbf{x}), y)\|_2 \leq K \|\mathbf{x}' - \mathbf{x}\|_p \leq \epsilon K,
$$

due to the definition of $\mathcal{S}(\mathbf{x})$. We also note that $\mathcal{J}(\mathbf{x}, y, \boldsymbol{\theta})$ is the maximal cross-entropy loss $\mathcal{L}$ within $\mathcal{S}(\mathbf{x})$, such that we have

$$
\|\nabla_{\boldsymbol{\theta}} \mathcal{J}(\mathbf{x}, y, \boldsymbol{\theta}) - \nabla_{\boldsymbol{\theta}} \mathcal{L}(f_{\boldsymbol{\theta}}(\mathbf{x}), y)\|_2 \leq \epsilon K.
\tag{B.2}
$$

Combining Eq. (B.1) and Eq. (B.2), we can obtain Eq. (5).

Note that the bound is tight since the all the equalities can be reached. □

**Remark 1.** *We note that a recent work (Liu et al., 2020a) gives a similar result on gradient stability. The difference is that they assume the loss function satisfies an additional Lipschitzian smoothness condition as*

$$
\|\nabla_{\boldsymbol{\theta}} \mathcal{L}(f_{\boldsymbol{\theta}_1}(\mathbf{x}), y) - \nabla_{\boldsymbol{\theta}} \mathcal{L}(f_{\boldsymbol{\theta}_2}(\mathbf{x}), y)\|_2 \leq K_{\boldsymbol{\theta}} \|\boldsymbol{\theta}_1 - \boldsymbol{\theta}_2\|_2,
$$

*where $K_{\boldsymbol{\theta}}$ is another constant. Then they prove that*

$$
\|\nabla_{\boldsymbol{\theta}} \mathcal{J}(\mathbf{x}, y, \boldsymbol{\theta}_1) - \nabla_{\boldsymbol{\theta}} \mathcal{J}(\mathbf{x}, y, \boldsymbol{\theta}_2)\|_2 \leq K_{\boldsymbol{\theta}} \|\boldsymbol{\theta}_1 - \boldsymbol{\theta}_2\|_2 + 2\epsilon K.
$$

*It can be noted that with this new assumption, we can simply obtain this result by Theorem 1. Therefore, Theorem 1 is a more general result of the previous one.*

### B.1    THEORETICAL ANALYSIS FOR TRADES

Note that TRADES adopts the KL divergence in its adversarial loss. The KL divergence is defined on two predicted probability distributions over all classes, as

$$
\mathcal{D}(f_{\boldsymbol{\theta}}(\mathbf{x}) \| f_{\boldsymbol{\theta}}(\mathbf{x}')) = \sum_{y \in \{1, \ldots, C\}} f_{\boldsymbol{\theta}}(\mathbf{x})_y \cdot \log \frac{f_{\boldsymbol{\theta}}(\mathbf{x})_y}{f_{\boldsymbol{\theta}}(\mathbf{x}')_y}.
$$

However, based on the local Lipschitz continuity assumption of the clean cross-entropy loss (which is only concerned with the predicted probability of the true class) in Eq. (4), we cannot derive a similar theoretical bound on the gradient stability of TRADES as in Eq. (5). Therefore, we need to make a different assumption on the KL divergence. For example, suppose the gradient of the KL divergence satisfies

$$
\|\nabla_{\boldsymbol{\theta}} \mathcal{D}(f_{\boldsymbol{\theta}}(\mathbf{x}) \| f_{\boldsymbol{\theta}}(\mathbf{x}'))\|_2 \leq K' \|\mathbf{x}' - \mathbf{x}\|_p,
$$

for any $\mathbf{x} \in \mathbb{R}^d$, $\mathbf{x}' \in \mathcal{S}(\mathbf{x})$, and any $\boldsymbol{\theta}$, where $K'$ is another constant. We denote the adversarial loss of TRADES as $\mathcal{J}'(\mathbf{x}, y, \boldsymbol{\theta})$, then we have

$$\|\nabla_{\boldsymbol{\theta}}\mathcal{J}'(\mathbf{x}, y, \boldsymbol{\theta}_1) - \nabla_{\boldsymbol{\theta}}\mathcal{J}'(\mathbf{x}, y, \boldsymbol{\theta}_1)\|_2$$
$$\leq \|\nabla_{\boldsymbol{\theta}}\mathcal{L}(f_{\boldsymbol{\theta}_1}(\mathbf{x}), y) - \nabla_{\boldsymbol{\theta}}\mathcal{L}(f_{\boldsymbol{\theta}_2}(\mathbf{x}), y)\|_2 +$$
$$\beta\|\nabla_{\boldsymbol{\theta}} \max_{\mathbf{x}'\in\mathcal{S}(\mathbf{x})} \mathcal{D}(f_{\boldsymbol{\theta}_1}(\mathbf{x})\|f_{\boldsymbol{\theta}_1}(\mathbf{x}'))\|_2 +$$
$$\beta\|\nabla_{\boldsymbol{\theta}} \max_{\mathbf{x}'\in\mathcal{S}(\mathbf{x})} \mathcal{D}(f_{\boldsymbol{\theta}_2}(\mathbf{x})\|f_{\boldsymbol{\theta}_2}(\mathbf{x}'))\|_2$$
$$\leq \|\nabla_{\boldsymbol{\theta}}\mathcal{L}(f_{\boldsymbol{\theta}_1}(\mathbf{x}), y) - \nabla_{\boldsymbol{\theta}}\mathcal{L}(f_{\boldsymbol{\theta}_2}(\mathbf{x}), y)\|_2 + 2\beta\epsilon K'.$$

Although we can derive a similar bound on gradient stability of TRADES, this bound is not directly comparable to Eq. (5) since we cannot find the relationship between $K$ and $K'$. However, our empirical analysis on gradient magnitude in Sec. 3.2 has shown that TRADES is dominated by the clean cross-entropy loss at the initial training epochs, thus the gradient stability of the TRADES loss will be similar to that of the clean cross-entropy loss, as also revealed in Fig. 4(a). Therefore, the gradient of TRADES would be relatively stable.

## C  FULL EXPERIMENTS ON ROBUST OVERFITTING

### C.1  ADDITIONAL EXPERIMENTS ON EXPLAINING ROBUST OVERFITTING

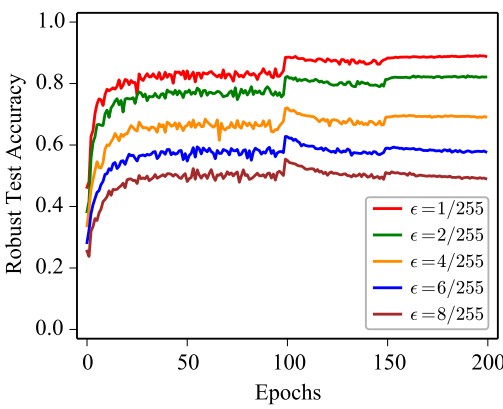
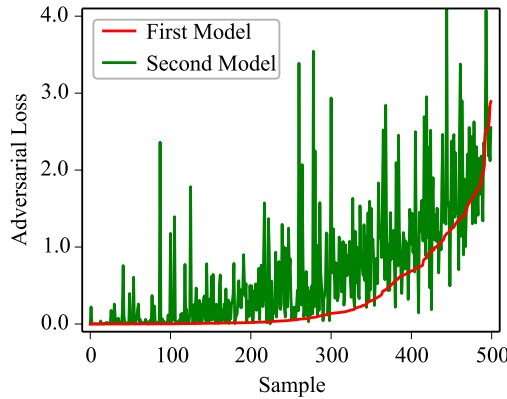

Figure C.1: The robust test accuracy of TRADES under various perturbation budgets $\epsilon$.

Figure C.2: The adversarial loss of WRN-28-10 and ResNet-18 trained by PGD-AT on 500 samples sorted by the loss of the first model (i.e., WRN-28-10).

First, we show the robust test accuracy curves of TRADES under various perturbation budgets in Fig. C.1. It can also be observed that when the perturbation budget is small, robust overfitting does not occur.

Second, we show that the "hard" training examples with higher adversarial loss values are consistent across different model architectures. We train one WRN-28-10 model and one ResNet-18 model based on PGD-AT. We then calculate the adversarial loss for each training sample for these two models. We show the adversarial loss on 500 samples sorted by the loss of the first model (i.e., WRN-28-10) in Fig. C.2. It can be seen that the samples with lower adversarial losses of the first model also have relatively lower losses of the second one and vice versa. The Kendall's rank coefficient of the adversarial loss between the two models is 0.78 in this case.

Third, we visualize the hard training examples with high adversarial loss values in Fig. C.3. It can be seen that these examples are difficult to recognize and their labels may be wrong. Therefore, the one-hot labels for these hard training examples can be noisy for AT, leading to the robust overfitting problem.

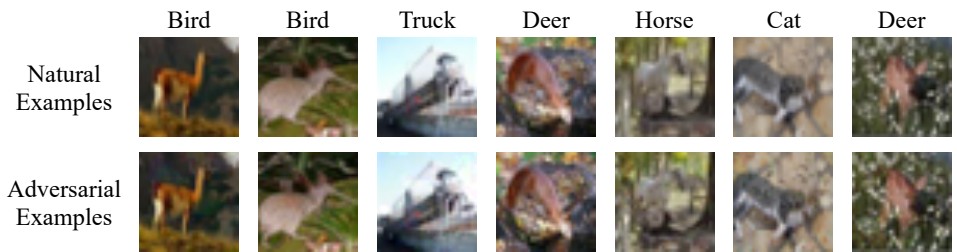

Figure C.3: The hard training examples with high adversarial loss values.

Table C.1: Test accuracy (%) of several methods using different model architectures and threat models. We choose the best checkpoint according to the highest robust accuracy on the test set under PGD-10.

| Methods | Networks | Norms | Natural Accuracy | | | PGD-10 | | |
|---|---|---|---|---|---|---|---|---|
| | | | Best | Final | Diff | Best | Final | Diff |
| PGD-AT | WRN-34-10 | | **86.58** | **86.83** | **-0.25** | 55.83 | 49.52 | 6.31 |
| PGD-AT**+TE** | WRN-34-10 | $\ell_\infty$ ($\epsilon = 8/255$) | 85.43 | 85.10 | 0.33 | **59.30** | **56.63** | **2.67** |
| PGD-AT | VGG-16 | | **79.60** | **81.26** | -1.66 | 48.52 | 43.02 | 5.50 |
| PGD-AT**+TE** | VGG-16 | | 78.19 | 79.13 | **-0.94** | **52.06** | **51.29** | **0.77** |
| PGD-AT | | | **88.82** | **88.96** | -0.14 | 69.05 | 65.96 | 3.09 |
| PGD-AT**+TE** | ResNet-18 | $\ell_2$ ($\epsilon = 128/255$) | 87.95 | 88.20 | -0.25 | **72.58** | **71.93** | **0.65** |
| TRADES | | | 86.50 | 86.57 | **-0.07** | 70.22 | 66.07 | 4.15 |
| TRADES**+TE** | | | **88.42** | **88.60** | -0.18 | **72.72** | **72.43** | **0.29** |

## C.2 ADDITIONAL EXPERIMENTS ON MITIGATING ROBUST OVERFITTING

We show the results of our proposed methods on other network architectures (including WRN-34-10 and VGG-16) and threat models (including $\ell_2$ norm) in Table C.1. The results consistently demonstrate the effectiveness of the proposed method.

We further show the results of PGD-AT, PGD-AT**+TE**, TRADES, and TRADES**+TE** on CIFAR-10 over 3 runs in Table C.2.

Table C.2: Test accuracy (%) of several methods on CIFAR-10 under the $\ell_\infty$ norm with $\epsilon = 8/255$ based on the ResNet-18 architecture. We show the mean/std of the results over 3 runs.

| Method | Natural Accuracy | | |
| --- | --- | --- | --- |
| | Best | Final | Diff |
| PGD-AT | **83.76** $\pm$ 0.02 | **84.93** $\pm$ 0.26 | -1.17 $\pm$ 0.29 |
| PGD-AT**+TE** | 82.36 $\pm$ 0.18 | 82.69 $\pm$ 0.14 | **-0.33** $\pm$ 0.31 |
| TRADES | 81.34 $\pm$ 0.15 | 82.70 $\pm$ 0.21 | -1.36 $\pm$ 0.36 |
| TRADES**+TE** | **83.66** $\pm$ 0.19 | **83.89** $\pm$ 0.09 | **-0.23** $\pm$ 0.21 |

| Method | PGD-10 | | |
| --- | --- | --- | --- |
| | Best | Final | Diff |
| PGD-AT | 52.62 $\pm$ 0.10 | 44.91 $\pm$ 0.01 | 7.71 $\pm$ 0.11 |
| PGD-AT**+TE** | **55.74** $\pm$ 0.17 | **54.82** $\pm$ 0.23 | **0.92** $\pm$ 0.07 |
| TRADES | 53.25 $\pm$ 0.07 | 50.48 $\pm$ 0.23 | 2.77 $\pm$ 0.16 |
| TRADES**+TE** | **54.93** $\pm$ 0.16 | **54.04** $\pm$ 0.19 | **0.89** $\pm$ 0.13 |

| Method | PGD-1000 | | |
| --- | --- | --- | --- |
| | Best | Final | Diff |
| PGD-AT | 51.26 $\pm$ 0.03 | 42.72 $\pm$ 0.06 | 8.54 $\pm$ 0.06 |
| PGD-AT**+TE** | **54.54** $\pm$ 0.27 | **53.01** $\pm$ 0.34 | **1.53** $\pm$ 0.24 |
| TRADES | 52.24 $\pm$ 0.20 | 48.74 $\pm$ 0.17 | 3.50 $\pm$ 0.13 |
| TRADES**+TE** | **53.55** $\pm$ 0.16 | **52.93** $\pm$ 0.07 | **0.62** $\pm$ 0.11 |

| Method | C&W-1000 | | |
| --- | --- | --- | --- |
| | Best | Final | Diff |
| PGD-AT | 50.24 $\pm$ 0.12 | 43.59 $\pm$ 0.07 | 6.65 $\pm$ 0.19 |
| PGD-AT**+TE** | **52.31** $\pm$ 0.01 | **51.67** $\pm$ 0.12 | **0.64** $\pm$ 0.11 |
| TRADES | 49.83 $\pm$ 0.05 | 48.11 $\pm$ 0.04 | 1.72 $\pm$ 0.02 |
| TRADES**+TE** | **50.80** $\pm$ 0.02 | **50.61** $\pm$ 0.07 | **0.19** $\pm$ 0.08 |

| Method | AutoAttack | | |
| --- | --- | --- | --- |
| | Best | Final | Diff |
| PGD-AT | 47.85 $\pm$ 0.17 | 41.62 $\pm$ 0.16 | 6.23 $\pm$ 0.26 |
| PGD-AT**+TE** | **50.37** $\pm$ 0.22 | **49.36** $\pm$ 0.24 | **1.01** $\pm$ 0.03 |
| TRADES | 48.86 $\pm$ 0.18 | 46.73 $\pm$ 0.07 | 2.13 $\pm$ 0.11 |
| TRADES**+TE** | **49.40** $\pm$ 0.27 | **48.77** $\pm$ 0.21 | **0.63** $\pm$ 0.05 |

