# OpenReview forum: "Exploring Memorization in Adversarial Training"
_ICLR.cc/2022/Conference — ICLR 2022 Poster_

### Official Review · Reviewer_c7wV · 2021-11-02

**Correctness:** 4
**Technical Novelty And Significance:** 3
**Empirical Novelty And Significance:** 4
**Recommendation:** 10
**Confidence:** 4

**Main Review:**

## Strengths
1. **Clean methodology and in-depth analysis**: I admire the effort made by the authors to provide a very in-depth analysis of their results, thoroughly checking different metrics, datatasets, and training setups, and trying to identify the root cause of all the described phenomena.
2. **Relevance of the results**: The fact that PGD-AT cannot memorize random labels due to its gradient instability, while TRADES or a warmed-up version of PGD-AT can, is quite significant. It points towards the importance of the early learning phase for adversarial training and it can spark new research avenues to understand the dynamics of adversarial training. Similarly, the fact that different complexity measures proposed in the literature to explain generalization do not correlate, and sometimes anti-correlate, with actual performance, might have been extended from prior results on standard training, but it is still of great relevance.
3. **Connection between robust overfitting and label noise**: The suggested cause for robust overfitting, i.e., introduction of label noise by the adversarial attacks, is bold, and the proposed method to overcome it seems quite effective.

## Weaknesses
1. **Theorem 1 does not say anything about stability of TRADES**: Although Theorem 1 strengthens the claim made by the authors explaining why PGD-AT has a greater gradient instability than standard training, it does not say anything on how TRADES can overcome it. As far as I understand it, a similar result might be true for the TRADES regularization term. This slightly weakens the theoretical arguments of the authors.
2. **Connection between robust overfitting and label noise could be more compelling**: As I said, I appreciate the bold claim made by the authors to explain the phenomenon of robust overfitting, and its connections to memorization in AT. However, I still believe some of their arguments are a bit speculative and could be better supported by the data. In particular, the fact that models share the ranking of difficulty of different examples is, in my opinion, a rather indirect way to measure the presence of label noise in a dataset. Something that would make the claim of the authors more convincing is to show that the "hard" instances they identified in the dataset do correspond to incorrectly classified or very noisy samples.
3. **Too many details off-loaded to appendix**: In my opinion, the authors postpone too many important discussions to the appendix which hurts the readability of the paper. Specifically, I believe that Fig. A8 and Appendix A2 should belong to the main body of text.

## Other comments
1. **Fine-tuning experiments**: I am of the opinion that the paper, in its current form, deserves to be accepted to the conference. However, I still believe there are certain experiments which, if included in the paper, would greatly strengthen the relevance of this work, and could lead me to increase my score. Specifically, there are two important works in the literature which study the effect of pre-training with random labels in performance (Liu et al. 2020, Maennel et al. 2020) and whose transferrability to the adversarial setting is currently unknown. Considering the fact that some of the experiments in this work point towards the importance of the early stages in training in the memorization behavior of AT, I wonder how different pre-training schemes with clean and random labels affect the findings of the authors. Namely, does temporal ensembling work better when the network is standardly trained on clean labels? Do neural networks robustly generalize when adversarially pretrained on random labels? Do they learn meaningful representations? I truly believe that answering any of these questions would be of great significance to the community, and therefore would make this work much more relevant.
   - Shengchao Liu, Dimitris Papailiopoulos, Dimitris Achlioptas. "Bad Global Minima Exist and SGD Can Reach Them." NeurIPS 2020
   - Hartmut Maennel, Ibrahim Alabdulmohsin, Ilya Tolstikhin, Robert J. N. Baldock, Olivier Bousquet, Sylvain Gelly, Daniel Keysers. "What Do Neural Networks Learn When Trained With Random Labels?". NeurIPS 2020
2. **Inconsistent x-axis in Fig. 4**: Fig.4a and Fig. 4b are technically showing complementary views of the instability of gradients in the PGD loss landscape. However, the x-axis in this plot is not consistent. One of these plots shows the change of gradient around a specific weight location, and the other the change of gradient during training. This looks slightly suspicious (although I feel it is minor) and I would encourage the authors to either use the same metric in both plots, or show the analogous result for the other x-axis in the appendix.



**Summary Of The Paper:**

This paper presents a thorough investigation of the dynamics of adversarial training (AT) when a network is trained on a dataset with random labels. This methodology was very successful at identifying theoretical gaps in our understanding of neural networks for standard training, so here the authors propose to replicate those experiments in the context of adversarial robustness. Surprisingly, the authors find that PGD-AT is incapable of fitting random labels when trained from scratch, while TRADES or a warmed-start version of PGD-AT can. They also suggest plausible explanations for this behavior. Finally, the authors use their novel findings to explore the reasons behind robust overfitting, and propose a new method to prevent it.


**Summary Of The Review:**

This paper presents a very rigorous empirical investigation of the dynamics of memorization in adversarial training. The provided insights are novel, properly isolated and clearly explained. Besides, the results are not only insightful from a theoretical perspective, but they also provide actionable insights. The inclusion of a few additional experiments could strengthen more the paper, but in the current form, I believe the paper is good enough for acceptance.

---

> ### Author Response · Authors · 2021-11-15
> **Thank you for the supportive review**
>
> Thank you for the supportive review. We have uploaded a revision of our paper. Below we address the detailed comments.
>
> ***Question 1: Stability of TRADES***
>
> Based on the local Lipschitz continuity assumption of the clean cross-entropy loss in Eq. (4), we cannot derive a similar theoretical bound on the gradient stability of TRADES as in Eq. (5), since the KL divergence in TRADES measures the discrepancy between two probability distributions of all classes while the cross-entropy loss is only concerned with the predicted probability of the true class. Although we can derive a similar bound by making a different assumption on the KL divergence, the bound is not directly comparable to Theorem 1, making it meaningless. However, our empirical analysis on gradient magnitude has shown that TRADES is dominated by the clean cross-entropy loss at the initial training epochs, thus the gradient stability of the TRADES loss will be similar to that of the clean cross-entropy loss, as also revealed in Fig. 4(a). Therefore, the gradient of TRADES would be relatively stable. In the revision, we provide more discussions on this issue in Appendix B.1.
>
>
>
> ***Question 2: Connection between robust overfitting and label noise***
>
> Thanks for the suggestion. In the revision, we visualize the hard training examples with high adversarial loss values in Fig. C.3. It can be seen that these examples are difficult to be recognized and their labels may be wrong. Therefore, the one-hot labels for these hard training examples can be noisy for AT, leading to the robust overfitting problem.
>
>
> ***Question 3: Too many details off-loaded to appendix***
>
> Thanks for the suggestion. In the revision, we move Fig. A.8 (now Fig. 5) to the main paper. Due to the strict page limitation, we cannot move all relevant details in Appendix to the main paper. We will further polish the paper in the final version.
>
> ***Question 4: Fine-tuning experiments***
>
> Thanks for pointing out the related works on studying the effect of pre-training with random labels.
> We further conduct experiments to explore different pre-training schemes in AT.
>
> First, we find that standard pre-training on clean labels does not affect the performance of different AT methods, including our proposed Temporal Ensembling approach. The possible reason is that the local optima of standard training and AT could be significantly different, and thus standard pre-training does not have any impact on AT.
>
> Second, we explore whether pre-training on random labels affects robust generalization in AT. Following Liu et al. (2020), we consider two initialization strategies — random initialization and adversarial initialization generated by training on random labelling of the training data. We also consider two training methods — vanilla SGD training and SOTA SGD training with data augmentation (random crops and flips), weight decay, and momentum. Our results are consistent with Liu et al. (2020), that the adversarial initialization can lead to worse performance than the random initialization with vanilla SGD, but with the regularization techniques, the models with different initializations converge to nearly the same test accuracy. We show the learning curves in Fig. A.10 in the revision.
>
> ***Question 5: Inconsistent x-axis in Fig. 4***
>
> Fig. 4(a) shows the change of gradients around a specific weight initialization $\theta$ following [1]. Then the x-axis $\lambda$ is the length moving $\theta$ to a random direction $\mathbf{d}$. The y-axis measures the $\ell_2$ distance between the gradients at $\theta$ and $\theta+\lambda\mathbf{d}$ to be consistent with Theorem 1. Fig. 4(b) shows the cosine similarity between the gradients during training. We did not use the $\ell_2$ distance since the gradient norm would change dramatically in the initial training stage. Thus the x-axis is the training epochs and the y-axis is the cosine similarity. Note that the x-axes of these two subplots cannot be the same, but the y-axes of these two subplots can be the same. In the revision, we re-plot Fig. 4(a) by using cosine similarity as the y-axis, as shown in Fig. A.8. The results based on cosine similarity are consistent with the results based on the $\ell_2$ distance, showing that the gradient of PGD-AT changes more abruptly.
>
> Reference:
>
> [1] Li et al. Visualizing the loss landscape of neural nets. NeurIPS 2018.

---

> > ### Comment · Reviewer_c7wV · 2021-11-19
> > **Answer to authors**
> >
> > Thank you very much for your detailed answer, and for making the effort to address all my concerns and implement my suggestions in the new manuscript.
> >
> > All in all, having read your comments, the other reviews, the updated version of the paper, and your answers to the reviews I strongly believe this is a very good paper: It is technically sound, it has a very thorough empirical evaluation, it deals with a timely topic, and clearly provides actionable insights. The concerns regarding novelty raised by other reviewers in my opinion are very minimal, and a degree of "disconnection" between theory and practice is of course expected when dealing with complex phenomena such as the one presented in this work. The theoretical result is still insightful and properly discussed in my opinion.
> >
> > Overall, **I believe this work should be highlighted in the conference**, so I will update my score appropriately.

---

> > > ### Author Response · Authors · 2021-11-19
> > > **Thanks for the update**
> > >
> > > Thank you very much for increasing the score and valuable comments. We'll try our best to further improve the paper in the final version.

---

### Official Review · Reviewer_g5ke · 2021-11-03

**Correctness:** 4
**Technical Novelty And Significance:** 3
**Empirical Novelty And Significance:** 3
**Recommendation:** 8
**Confidence:** 2

**Main Review:**

**Abstract and introduction:**

The abstract and introduction are very well written and clear. This is a big strength. Figure 1 is also very clear.

**Section 3.1:**

The fact that PGD-AT fails to converge and differs so much from TRADES is not intuitive and a great contribution. The fact that this happens over several datasets, architectures, and threat models does indeed imply that the adversarial training algorithm has a significant impact on the convergence and not the capacity of the network.

**Section 3.2:**

This section is very intuitive and it makes sense that the authors consider the gradient norm and stability. It makes sense that the instability of the gradient would cause PGD-AT to fail to converge, However, even though the upper bound in Theorem 1 is tight, it can still be far off for specific values of the inputs. So the theoretic justification that the gradient of the PGD-AT loss will vary drastically based on this upper bound is not very strong.

**Section 4.1:**

The author’s hypothesis that memorization of one-hot labels causes robust overfitting is convincing and they provide evidence of this. This is a strength.

**Experiments:**

Overall one of the biggest strengths of this paper is the thoroughness of the experiments. The authors did a great job of providing a significant amount of evidence for their claims.


**Summary Of The Paper:**

Memorization has been investigated in deep neural network classifiers but not that much in adversarial training. The authors investigate the ability of adversarial training to memorize random datasets. They use PGD-AT and TRADES to do the adversarial training and see different convergence properties, which differs significantly from training on true labels. They also discover that robust overfitting can occur in adversarial training with memorization and use temporal ensembling to mitigate it.

**Summary Of The Review:**

Overall, the paper is very well written, what the authors do is clear, makes sense, and is supported by a significant amount of data. There are really few weaknesses with the paper.

---

> ### Author Response · Authors · 2021-11-15
> **Thank you for the supportive review**
>
> Thank you for the supportive review. We have uploaded a revision of our paper.
>
> ***Question: Theoretical justification of gradient stability***
>
> We agree that the tight bound in Theorem 1 can be hardly achieved in practice. Thus we conducted an experiment to verify the gradient instability issue of PGD-AT, as shown in Fig. 4. It can be seen that the gradient of PGD-AT changes abruptly while the gradients of TRADES and the clean CE loss are more continuous. The empirical results are consistent with our theoretical analysis.

---

### Official Review · Reviewer_x9Ej · 2021-11-03

**Correctness:** 3
**Technical Novelty And Significance:** 2
**Empirical Novelty And Significance:** 2
**Recommendation:** 3
**Confidence:** 4

**Main Review:**

Overall, the paper presents several interesting insights about the behavior of adversarial training algorithms. And the proposed methods are shown to out-perform PGD-AT and TRADES. However, the findings in the paper are disconnected and some of them lack explanations. Therefore, it is hard to fully understand the importance of these findings. Moreover, the proposed method, the Temporal Ensembling (TE) approach, does not have significant novelty compared to previous methods. Next, we detail our concerns.

1. In Section 3.1 and Section 3.2, the authors make great efforts to compare the difference between algorithmic stability of PGD-Train and TRADES, when 100% training labels are flipped. However, the stability difference will not result in any impact to the models, under the cases if fewer labels (0-80%) are flipped (Figure 2). Therefore, it is hard to justify the contribution of this analysis, if this difference is always oblivious under general scenarios.
2. In Section 3.3, the authors also mention that currently existing complexity measures cannot adequately explain the robust generalization performance of AT. What is the reason for the gap between theories and the practices?
3. Moreover, in Section 4.1, the authors deploy experiments to demonstrate that there are “hard” training samples which are consistently hard to be classified. However, why the existence of hard training examples can support the claim they can result in overfitting issues? What if removing these “hard” / “noisy” samples from the training set, will this resolve the overfitting problem?
4. For the proposed method, the Temporal Ensembling (TE) approach, it is not very different from an existing work [1]. Although the Equ.(7) in this paper and Algorithm 1 in [1] only have minor differences, both the papers are designed to (adversarially) train one model, with a self-adapted label instead of the one-hot labels. Furthermore, these two papers have the same way to generate the self-adapted labels. The only difference is the training objective. However, they also have high similarity in terms of definitions and experimental effects.

**References**:
[1] Huang, Lang, Chao Zhang, and Hongyang Zhang. "Self-adaptive training: beyond empirical risk minimization." Advances in Neural Information Processing Systems 33 (2020).


**Summary Of The Paper:**

This paper provides comprehensive studies on several facts about the memorization of adversarial training algorithms (AT). Specifically, the authors uncover two findings: (1) Given a sufficient model capacity, PGD Adversarial Training cannot fit the whole training set (with 100% labels flipped). While, TRADES can fit the whole random flipped set. It may be because the objective function of TRADES has a clean loss, which can improve the stability of training. (2) Memorization of one-hot labels in AT methods can be one important factor to the robust overfitting issues. Based on these findings, the authors propose a method Temporal Ensembling (TE) approach to avoid fitting all adversarial examples with one-hot labels.

**Summary Of The Review:**

Overall, the findings of this paper lack some clarifications about their significance. Moreover, the proposed adversarial training method does not have enough novelty beyond currently existing methods.

---

> ### Author Response · Authors · 2021-11-15
> **Thank you for the valuable review**
>
> Thank you for the valuable review. We have uploaded a revision of our paper.
>
> In this paper, we concentrate on the memorization effect in AT. We first show the empirical results of memorization with random labels (Sec. 3.1) and then discuss the connections to the convergence (Sec. 3.2) and generalization (Sec. 3.3) of AT. The memorization analysis can help to explain and mitigate the robust overfitting problem (Sec. 4). We make a great effort to provide in-depth analyses of different aspects related to memorization in AT with significant results, as agreed by other reviewers.
>
> Below we address the detailed comments.
>
> ***Question 1: Contribution of the stability analysis***
>
> The contributions of the stability analysis in Sec. 3.1 and Sec. 3.2 are two-fold. First, as we discovered an uncommon phenomenon that PGD-AT and TRADES demonstrate different behaviors when trained on random labels, it is necessary to perform a convergence/stability analysis to understand the phenomenon in depth. The analysis provides new insights on the differences between PGD-AT and TRADES, which are not studied previously.
> Second, we need to clarify that our analysis indeed reveals the stability of PGD-AT in more realistic scenarios. As seen in Fig. 2(c), PGD-AT cannot converge with 80% noise rate, while TRADES can converge. We provided an empirical analysis based on gradient stability in Appendix A.2.3. Besides, when the perturbation budget $\epsilon$ is large (e.g., $\epsilon=64/255$), PGD-AT cannot converge even with true labels, but TRADES can achieve about $50\%$ training accuracy. This can also be explained by our convergence analysis that the gradient is very unstable in PGD-AT with a larger perturbation budget, making it fail to converge.
>
> ***Question 2: The gap between theory and practice for robust generalization***
>
> The complexity measures we studied in Sec. 3.3 include both theoretically and empirically motivated ones. The norm-based measures are derived from the theoretical generalization bound [1], which makes various assumptions, e.g., the neural networks should be two-layer feedforward networks with bounded norm of the weight matrices. These assumptions can be hardly held in practice, such that the suggested norm-based measures are inadequate explain robust generalization, as also discussed in standard training [2]. The sharpness/flatness-based measures are proposed by empirical analyses, which have been shown to be well correlated with the robust generalization gap by only considering models trained on true labels. Our work further considers models trained on random labels and points out the insufficiency of these measures in this case.
>
> ***Question 3: The connection between hard training samples and robust overfitting***
>
> As discussed in Sec. 4.1, the one-hot labels of the hard training samples can be noisy for AT. In most AT methods, the network is forced to fit/memorize one-hot labels, which can lead to robust overfitting due to memorization of these hard examples. We indeed tried to remove these noisy samples in the dataset to mitigate robust overfitting (see Footnote 2 in Page 7 in the original version), but we found that the performance is worse than our method due to the reduction of training data. In the revision, we move the result to Table 2 to make it clearer.
>
> ***Question 4: The novelty of the proposed Temporal Ensembling approach***
>
> Since the discovery of robust overfitting, there still lacks a reasonable explanation of why it occurs. In Sec. 4, we draw a connection between memorization and robust overfitting, showing that robust overfitting is caused by memorizing one-hot labels, which could be noisy for some data. To solve the discovered problem, we propose a simple strategy by integrating the Temporal Ensembling approach into AT, which is typical and effective to overcome label noise. Therefore, the main novelty is that we identify the problem to cause robust overfitting and propose a simple method to address it.
> Although the self-adaptive training (SAT) method also replaces one-hot labels by the self-adapted labels, it can be viewed as a different solution to our identified problem. Technically, SAT still uses the cross-entropy loss with the self-adapted labels, but our method adds a regularization term to the original cross-entropy loss to enforce the smoothness of the outputs. The empirical results in Table 2 also show the effectiveness of our approach upon SAT.
>
> Reference:
>
> [1] Yin et al. Rademacher complexity for adversarially robust generalization. ICML 2018.
>
> [2] Jiang et al. Fantastic generalization measures and where to find them. ICLR 2020.

---

> > ### Comment · Reviewer_x9Ej · 2021-11-24
> > **Reply to the rebuttal.**
> >
> > We thank the author to answer our questions. However, we still have the concerns.
> >
> > **Towards Answer of Question 1: Contribution of the stability analysis**
> >
> > It is hard to agree that the analysis has significant importance in realistic scenarios. The convergence difference between AT and TRADES only happens when there is 80% nosily label rate, or very larger perturbation budget 64/255. It is not realistic or practical. For example, people will never train a model with 80% noisy labels. Similarly, they will never do adversarial training / TRADES with such high perturbation budget like 64/255. Until now, SOTA methods cannot even achieve high robustness using adversarial training under 8/255 in CIFAR10.
> >
> > **Towards Answer of Question 2: The gap between theory and practice for robust generalization**
> >
> > As the author mentioned, “the previous theoretical papers focus on the neural networks with two-layer feedforward networks with bounded norm of the weight matrices. These assumptions can be hardly held in practice.” So why does the author think it is surprising (or interesting) that their theoretical bounds cannot align well with the generalization gap in practice for deep models?
> >
> > We suppose the authors try to claim that: the (robustness) generalization for DNN models cannot be well-explained by classic generalization theories which are based on complexity & flatness measurements. However, given that previous studies focused on natural training already figure this point out [1], it is not surprising that the similar phenomenon happen in robust training algorithms. Moreover, the analysis in this part lacks deep discussions. Thus, it is extremely difficult to comprehend the importance of the analysis.
> >
> > **Towards Answer of Question 4: The novelty of the proposed Temporal Ensembling approach**
> >
> > We agree with the author that the specific formulations of TE and SAT are different. However, they have similar high-level ideas and effects. For example, in terms of “soft-label” ($p_i$) design, they are exactly the same:
> > $$\text{SAT:  } p_i = \eta \cdot p_i + (1-\eta)(f(x_i))$$
> > $$\text{TE:  } p_i = \eta \cdot p_i + (1-\eta) (f(x_i))$$
> >
> > in terms of model training objective:
> > $$\text{SAT:  } L = CE(f(x_i), p_i)$$
> > $$\text{TE:  } L = CE((f(x_i), y_i) + ||f(x_i) - p_i||_2^2$$
> > They have similar functions to encourage the model outputs scores which align well with the soft-label $p_i$.
> >
> > Moreover, from the empirical results, SAT and PGD-AT+TE are also very similar. For example, from Table 2 in the paper, the difference between their clean accuracy & robust accuracy (via AA attack) is smaller than 1%.
> >
> > **Summary**
> >
> > As the author didn’t properly address my concerns, currently I maintain my score. Moreover, the novelty of the method is my major concern.
> >
> > **Reference**
> >
> > [1] DEEP DOUBLE DESCENT: WHERE BIGGER MODELS AND MORE DATA HURT, Nakkiran et al, 2019

---

> > > ### Author Response · Authors · 2021-11-25
> > > **Thank you for the further comments**
> > >
> > > ***Further response to Question 1: Contribution of the stability analysis***
> > >
> > > It is common practice to consider 80% label noise in the research field of (standard) training with noisy labels. Improving model robustness under a larger perturbation budget has also been studied in recent work [1,2]. Although most AT methods consider the general setting with $\epsilon=8/255$ currently, it does not mean that studying other settings (such as AT with 80\% noisy labels or a larger perturbation budget) is not realistic nor important. Future work may study AT under the more challenging settings and find useful analysis in our work.
> > >
> > > ***Further response to Question 2: The gap between theory and practice for robust generalization***
> > >
> > > The main contribution of the generalization analysis is to show the recently suggested complexity measures (including norm-based and flatness-based measures) cannot adequately explain robust generalization by considering models trained on random labels. Although this has been extensively studied in standard training, there still lacks a similar analysis in AT, which we think is complementary and important to the understanding of generalization. Besides, previous work also adopted these complexity measures to improve model robustness, e.g., [3] proposed to use $\ell_1$ regularization in AT. We found that there is no causal relationship between the complexity measure and robust generalization, which also challenges the rationality of previous work.
> > >
> > > ***Further response to Question 4: The novelty of the proposed Temporal Ensembling approach***
> > >
> > > As we clarified in the initial response, the major novelty of the robust overfitting analysis is identifying the problem to cause robust overfitting -- the one-hot labels used in AT are noisy. We simply apply the typical Temporal Emsembling approach to address the identified problem. Other advanced methods to learn robustly under label noise may also be applied to alleviate robust overfitting, including SAT. The similar formulations of SAT and TE can only show that SAT does not have enough novelty upon TE.
> > >
> > > Reference:
> > >
> > > [1] Stutz et al. Confidence-calibrated adversarial training: Generalizing to unseen attacks. ICML 2020.
> > >
> > > [2] Shaeiri et al. Towards deep learning models resistant to large perturbations. arXiv 2020.
> > >
> > > [3] Yin et al. Rademacher complexity for adversarially robust generalization. ICML 2018.

---

### Official Review · Reviewer_erPY · 2021-11-10

**Correctness:** 3
**Technical Novelty And Significance:** 3
**Empirical Novelty And Significance:** 2
**Recommendation:** 6
**Confidence:** 4

**Main Review:**

Strength
This paper is easy to follow and well written and the hypothesis regarding the difference between PGD and TRADES is well analyzed by separating the gradient of each loss to provide an easy-to-understand analysis. To validate its observation, the authors designed a set of comprehensive experiments for evaluation and comparison purposes. This paper also proposes a new hypothesis about the cause of robust overfitting, which can be intuitively understood and supported by comprehensive experiments.  Also in section 5, It is experimentally well designed and confirmed that the proposed method by the regularization term using the temporal ensembling approach alleviates robust overfitting, and showed better robust accuracy than the existing methods.

Weakness
1) Complexity measures : In Figure 5, the authors show that the previous complexity measures cannot adequately explain and ensure the robust generalization performance in AT. However, since there are few model plots shown in the graph, there seems to be insufficient justification that the previous approaches are not suitable. The spectral norm and flatness-based measure in Fig. 5 may be able to explain the generalization gap if the scale is different and more model plots are expressed. It would be nice to alleviate the expression that all these measures are inadequate. It would be better if more abundant model settings and experimental results were presented. Moreover, there is a lack of explanation of why previous measures are not adequate in this setting and it would be better if the authors could clearly suggest a new alternative.
2) Robust overfitting : Although it has been shown empirically that overfitting is improved through the Temporal Ensembling (TE) approach, there is a lack of novelty for the method itself. The proof of hypothesis by Fig 6 leaves little doubt, since the phenomenon in Fig 6(b) is natural because robust overfitting will not occur if we do not use AT. Also in Fig 6(c), the phenomenon can be due to adversarial transferability[1].
3) Overall, It would be nice if all experiments were configured in mean/std format in various initial parameters (e.g., random seed) of the model.
4) In section 4.2, authors claimed that most of the approaches for noisy labels are not suitable for AT, since excluding the noisy label leads to the reduction of the training data. However, these approaches already showed good performance with small datasets like MNIST and CIFAR. It is hard to understand why this suddenly became a problem, even all natural examples can be treated as ‘clean examples’ in AT. If the lack of training data is a main problem, will it be automatically relieved when we use a bigger dataset, like ImageNet?

5)  An adversarial example is a variable of model parameters, and it is continuously changed as training proceeds. Is it appropriate to apply a Temporal Ensemble to this problem, which assumes that the training data is unchanged and takes a moving average of prediction for several epochs?

6) In Theorem 1, the authors claimed that the upper bound of the inequality is tight, because the equality can be reached. However, in order for the inequality to be equal, four gradients - adversarial example with theta 1, clean example with theta 1, clean example with theta 2, adversarial example with theta 2 - must exist in a straight line within a millions of dimensions of space. And the equality of Lipschitz constraints should be reached for both points. It is hard to believe that this can be occured in a real experiment. Furthermore, even assuming that the equality can be reached, it does not provide any information about the magnitude of the expected gap in the inequality. Experimental evidence is needed to continue to assert this.

7) Apart from that, the authors claimed that the right side of the inequality is large due to the Lipschitz constant K. But the Lipschitz continuity of the norm of the gradient is not a widely used concept, and the typical value of Lipschitz constant K in deep neural networks may not be common sense. It would be better if there is a justification about why 2*epsilon*K is (obviously) bigger than the difference between the gradients.

[1] Adversarial Examples Are Not Bugs, They Are Features (https://arxiv.org/abs/1905.02175)


**Summary Of The Paper:**

This paper presents the memorization effect/behavior in adversarial training in perspective of model capacity, convergence and generalization. They first investigate the difference of memorization behavior between two adversarial training methods, PGD-AT and TRADES, with random labels and provide the proof of gradient instability for PGD-AT in terms of Lipschitz constant. Also they suggest that previous complexity measures are inadequate to explain robust generalization. Lastly, they investigate the cause of robust overfitting in AT and propose a mitigation algorithm by a regularization term for avoiding the excessive memorization of adversarial examples.

**Summary Of The Review:**

This paper is easy to follow and well written and the hypothesis regarding the difference between PGD and TRADES is well analyzed. They present the problem of memorization in AT by robust overfitting and propose the regularization term by temporal ensembling approach with their hypothesis, also show the effectiveness of the proposed method in systematic experiments.

---

> ### Author Response · Authors · 2021-11-15
> **Thank you for the valuable review (Part 2/2)**
>
> ***Question 4: Excluding the noisy labels***
>
> Although the methods for excluding noisy labels show good results for standard training on CIFAR, they can lead to inferior results for AT since the reduction of training samples can affect the performance of AT [3]. To show this, we provide the results of Co-teaching [4] (adapted to PGD-AT) in Table 2, which achieves worse performance compared with our method, though robust overfitting is alleviated. Using a larger dataset can relieve this issue [5,6,7].
>
> ***Question 5: Moving average of predictions in Temporal Ensembling***
>
> We actually calculate the moving average of predictions of natural examples instead of adversarial examples (see the first paragraph in Page 8).
>
> ***Question 6: Experimental evidence of Theorem 1***
>
> We agree that the tight bound in Theorem 1 can be hardly achieved in practice. We indeed provided the experimental result in Fig. 4(a), which empirically shows that the gradient of PGD-AT changes abruptly, supporting the theoretical result of Theorem 1.
>
> ***Question 7: Justification why $2\epsilon K$ is bigger***
>
> Assume that the difference between $\theta_1$ and $\theta_2$ approaches 0, the $\ell_2$ distance between the gradients of the clean cross-entropy loss will also approach 0 if the gradient satisfies another Lipschitz continuity condition on $\theta$, which is empirically verified in Fig. 4(a). But $2\epsilon K$ is a constant, which can be much larger than the distance between the gradients.
>
> Reference:
>
> [1] Yin et al. Rademacher complexity for adversarially robust generalization. ICML 2018.
>
> [2] Jiang et al. Fantastic generalization measures and where to find them. ICLR 2020.
>
> [3] Schmidt et al. Adversarially robust generalization requires more data. NeurIPS 2018.
>
> [4] Han et al. Co-teaching: Robust training of deep neural networks with extremely noisy labels. NeurIPS 2018.
>
> [5] Hendrycks et al. Using pre-training can improve model robustness and uncertainty. ICML 2019.
>
> [6] Alayrac et al. Are labels required for improving adversarial robustness? NeurIPS 2019.
>
> [7] Carmon et al. Unlabeled data improves adversarial robustness. NeurIPS 2019.

---

> ### Author Response · Authors · 2021-11-15
> **Thank you for the valuable review (Part 1/2)**
>
> Thank you for the valuable review. We have uploaded a revision of our paper. Below we address the detailed comments.
>
> ***Question 1: Complexity measures***
>
> Although we only used 12 models, the results shown in Fig. 5 (Fig. 6 in the revision) demonstrate that some complexity measures are anti-correlated with the robust generalization gap, indicating that they cannot explain robust generalization. We agree that the flatness of the loss landscape is a more reliable complexity measure, and we have revised our argument in the revision.
>
> The complexity measures we studied in Sec. 3.3 include both theoretically and empirically motivated ones. The norm-based measures are derived from the theoretical generalization bound [1], which makes various assumptions, e.g., the neural networks should be two-layer feedforward networks with bounded norm of the weight matrices. These assumptions can be hardly held in practice, such that the suggested norm-based measures are inadequate to explain robust generalization, as also discussed in standard training [2]. The sharpness/flatness-based measures are proposed by empirical analyses, which have been shown to be well correlated with the robust generalization gap by only considering models trained on true labels. Our work further considers models trained on random labels and points out the insufficiency of these measures in this case, although the flatness of the loss landscape is the most adequate one among all measures we studied. Our analysis can motivate further research on developing more appropriate complexity measures of robust generalization, which we leave to future work.
>
> ***Question 2: Robust overfitting***
>
> **Novelty.** Since the discovery of robust overfitting, there still lacks a reasonable explanation of why it occurs. In Sec. 4, we draw a connection between memorization and robust overfitting, showing that robust overfitting is caused by memorizing one-hot labels, which could be noisy for some data. To solve the discovered problem, we propose a simple strategy by integrating the Temporal Ensembling approach into AT, which is typical  and effective to overcome label noise. Therefore, the main novelty is that we identify the problem to cause robust overfitting and propose a simple method to address it.
>
> **Fig. 6.** (Fig. 7 in the revision) We agree that robust overfitting will not occur without AT (i.e., $\epsilon=0$). But what we find in Fig. 6(b) is that robust overfitting becomes more obvious when the perturbation budget $\epsilon$ gets larger. It is due to that when the perturbation budget is larger, the one-hot labels are noisier for adversarial examples in AT, making robust overfitting more obvious. This phenomenon can well support our argument that robust overfitting is caused by memorizing (noisy) one-hot labels.
> In Fig. 6(c), the consistency between hard examples may be viewed as another kind of transferability across models. Due to the consistency/transferability, we validate that these hard examples are intrinsic of a dataset, supporting our hypothesize on why robust overfitting occurs.
>
> ***Question 3: Multiple runs***
>
> Thanks for the suggestion. In the revision, we conduct the same experiments as Table 1(a) and show the results over 3 runs in Table C.2. We will provide more results over multiple runs in the final version.

---

### Decision · Program_Chairs · 2022-01-20

**Decision:**

Accept (Poster)

**Comment:**

This paper demonstrates that deep networks can memorize adversarial examples of training data with completely random labels, which motivates some analyses on the convergence and generalization of adversarial training (AT). The authors identify a significant drawback of memorization in AT that could result in robust overfitting and propose a new algorithm to mitigate this drawback. Experiments on benchmark datasets validate the effectiveness of the proposed algorithm. One of the reviewers is concerned about (1) the validity of stability analysis where 80% of the data labels are noisy, and the perturbation (64/255) is large, (2) the gap between theory and practice, and (3) novelty. The authors have made a great effort to address these concerns. Although there is still no consensus after the author's response, the majority of the reviewers are in strong support. I, therefore, recommend acceptance.